# Visual scanning strategies in the cockpit are modulated by pilots' expertise: A flight simulator study

**Christophe Lounis**(ORCID)*, **Vsevolod Peysakhovich, Mickaël Causse**

ISAE-SUPAERO, Université de Toulouse, Toulouse, France

* christophe.lounis@isae-supaero.fr

**Data Availability Statement:** All relevant data are within the manuscript and its Supporting information files.

**Funding:** This research was supported by a chair grant from Dassault Aviation (\CASAC", holder:

## Abstract

During a flight, pilots must rigorously monitor their flight instruments since it is one of the critical activities that contribute to update their situation awareness. The monitoring is cognitively demanding, but is necessary for timely intervention in the event of a parameter deviation. Many studies have shown that a large part of commercial aviation accidents involved poor cockpit monitoring from the crew. Research in eye-tracking has developed numerous metrics to examine visual strategies in fields such as art viewing, sports, chess, reading, aviation, and space. In this article, we propose to use both basic and advanced eye metrics to study visual information acquisition, gaze dispersion, and gaze patterning among novices and pilots. The experiment involved a group of sixteen certified professional pilots and a group of sixteen novice during a manual landing task scenario performed in a flight simulator. The two groups landed three times with different levels of difficulty (manipulated via a double task paradigm). Compared to novices, professional pilots had a higher perceptual efficiency (more numerous and shorter dwells), a better distribution of attention, an ambient mode of visual attention, and more complex and elaborate visual scanning patterns. We classified pilot's profiles (novices—experts) by machine learning based on Cosine KNN (K-Nearest Neighbors) using transition matrices. Several eye metrics were also sensitive to the landing difficulty. Our results can benefit the aviation domain by helping to assess the monitoring performance of the crews, improve initial and recurrent training and ultimately reduce incidents, and accidents due to human error.

## Introduction

### Monitoring activity in the cockpit

Throughout the flight, pilots must build and update their situation awareness (SA) to maintain flight safety margins [1]. The flight crew cannot update the SA without monitoring specific flight instruments (e.g., attitude indicator, speed, altimeter, engine parameters) and the external environment (by clear weather). The monitoring activity, particularly critical during dynamic flight phases such as take-off and landing, includes the observation and interpretation

MC). The funders had no role in study design, data collection and analysis, decision to publish, or preparation of the manuscript.

**Competing interests:** The authors have declared that no competing interests exist.

of the flight path data, aircraft-configuration status, automation modes, and on-board systems. It supposes a real-time comparison of instrument data or system modes against the expected values according to the current flight phase. A rigorous cockpit monitoring allows timely corrective actions in case of a parameter deviation, ensuring an optimal level of safety [2]. This monitoring activity is structured in sequences of attentional shifts from an instrument to another.

Irregularities in these sequences can undermine the safety margins. In numerous cases of aircraft accidents, pilots' visual scanning has been described as "inadequate", "ineffective", or "insufficient" [3]. Since the 1994 report by the National Transportation Safety Board that the inappropriate monitoring was involved in 84% of major accidents in the United States [4], numerous studies investigated the visual behavior of the pilots. However, in a "practical guide for improving flight path monitoring" by the Flight Safety Foundation [5], which investigated 188 accidents with monitoring issues, it is underlined that many monitoring errors still occur, most of them during dynamic phases of flight (e.g., climb, descent, approach, and landing). In 2013, the Federal Aviation Administration required airlines to include an explicit training program to improve monitoring skills [6, 7]. Following the PARG study [7], the Bureau d'Enquêtes et d'Analyses (French Investigation Agency) encouraged the use of eye tracking systems to finely analyze and improve crews' visual scanning. Interestingly, an extensive survey conducted on 931 pilots during the PARG study [8] showed that most of the pilots need a better description of what a "standard" visual circuit in the cockpit is. Similarly, in another recent survey [9], 75% of pilots deemed helpful to know the required visual patterns for the different flight phases to enhance their cockpit monitoring skills.

## Visual scanning strategies as a marker of expertise

The relationship between visual scanning skills and performance has been highlighted in experiences where participants were trained to gaze at relevant areas. For instance, Shapiro et al. [10] demonstrated that videogamers that were trained using efficient visual scanning examples showed better performance compared with random pattern training or no training at all. In another air traffic control study, Kang and Landry [11] enhanced novices' performance in a conflict detection task by presenting experts' visual scans overlaid on the radar screen during the task. The study also showed that the visual presentation outperformed the "instruction-only" condition. These studies support the relationship between visual patterns and task performance, and demonstrate the possibility to improve these patterns with adequate training. The task performance increases with the experience and associated expertise. The links between the visual scanning strategies and the expertise were observed in fields such as radiology, driving, sport, military aviation or chess (e.g., [12–14]). Gegenfurtner, Lehtinen, and Säljö [15] conducted a meta-analysis and highlighted that experts (compared to non-experts) generally demonstrate more fixations on task-relevant areas as well as shorter fixations. In their review of eye movements in medicine and chess, Reingold and Sheridan [16] have labeled this greater perceptual effectiveness of experts as "*superior perceptual encoding of domain-related patterns*".

Several studies in the aeronautical domain showed that pilots' visual scanning strategies (e.g. duration and frequency of fixations) evolve with the level of expertise [17–25]. According to Bellenkes et al. [26] the fixations of experts are shorter and fixations on instruments are more frequent. Similarly, Kasarskis, Stehwien, and Hickox [27] noticed that expert pilots (1500—2150 flight hours) perform more fixations and have shorter dwell times than novices (40—70 flight hours), and argued that experts have more structured visual patterns. Lorenz et al. [28] have shown that experts (3000–10300 flight hours) spend more time looking outside

the cockpit compared to novices (13–500 flight hours) during a taxiing task. Furthermore, a study involving fighter pilots flying high speed low altitude flights [29] highlighted the importance of efficient visual scanning strategies. In this study, the pilots who achieved the best flight performance made shorter fixations on the heads-down tactical display and alternated more frequently between the tactical display and the outside world. Similar results were found in experts (>1000 hours) and novices (200—400 hours) playing flight simulation games [30]. Because visual scanning appears to differentiate between expert and novice pilots' performance, it is interesting to examine which eye tracking metrics are available in the literature [31] to compare the visual scanning strategies using various approaches such as the estimation of the distribution and patterning of the visual scanning.

The objective of the present work is to provide a framework for eye movement data analysis techniques to study visual scanning strategies in novices and experts. These eye movement measures and algorithms are presented in light of the results of an experiment involving novice and expert pilots during a landing scenario performed in a flight simulator. We examined the impact of expertise and the difficulty of the flight scenario on the visual attention allocation. The participants performed three times the same landing scenario with varying difficulty conditions. Two difficulty conditions incorporated a supplementary visual monitoring task, with different time pressure, to make cockpit monitoring more complex by increasing visuomotor activity. We analyzed the effect of the pilots' profile (pilot vs. novice) as well as the effects of the landing difficulty on numerous standard (number of dwells, average dwell times) and advanced eye movements metrics (Lempel-Ziv Complexity, Gaze Transition Entropy, attentional modes, N-gram methods) presented in the following section.

## State-of-the-art visual scanning metrics

Classical eye movements measures such as fixation duration, dwell time, or the number of fixations, provide relevant results when comparing novices vs. experts. However, statistical analyses of these metrics often involve time-averaging operations, thus, neglecting the information regarding the sequence of instrument scanning. Consequently, a rich part of the data that reflects the dynamic of the deployment of the attention processes is lost or not fully exploited. Numerous other metrics are available to explore and characterize in more depth visual scanning strategies. We use the broad term "visual scanning" to describe visual scanning made up of an at least one dwell to one area of interest (AOI), followed by a transition, and a dwell to another AOI; "visual scanning pattern" is used when the visual scanning is made up of repeated sequences of a given "visual scanning". One approach to examine visual scanning strategies is to analyze transition matrix (e.g., [32–35], a second one is the characterization of fluctuation between ambient/focal visual behavior [36], another one is to derive global patterns metrics such as entropy (e.g., [37, 38], see [39] for a review). More generally, in this paper, we classified visual scanning strategies metrics in three AOI based approaches: one is based on Markov chains (transition matrix), another is based on the attentional modes, and the last one is based on sequences analyses. Fig 1 presents a comparison of the visual scanning metrics described below (e.g. formula, definition, strength shortcomings, strength, etc. . .).

### Markov chains

Several metrics allow examining whether visual scanning is narrow or wide.

**The transition matrix probabilities.** They contain the information about how often a transition from one Area Of Interest (AOI) to another occurred based on subsequent dwells of the visual scan. This method provides a data representation that can also lead to the development of stochastic and queuing models [40] of the pilot's scanning in the cockpit. This method

| AOI based approaches | Visual scanning metrics | Formula | Definition | Strength | Shortcoming |
|---|---|---|---|---|---|
| Markov chains | Transition Probability Matrix (TPM) | $P_{ij} = \frac{n_{ij}}{\sum_j n_{ij}}$ | Transition frequency from one AOI to another one based on X series of dwells. | Highlight strong link between AOI. | When more than 3 AOIs are displayed, it is particularly difficult to visualize. |
| | Transition Density Matrix (TDM) | $TDM = \frac{\sum_{I=1}^{N} \sum_{j=1}^{n} c_{i,j}}{n^2}$ | Ratio of non-null transitions cells divided by the total number. | Useful to compare participants when some AOIs are not observed. | Not useful if all AOIs are take into account by participants. |
| Attentional modes | Modified K-Coefficient | $K_i = \frac{d_i - \mu_d}{\rho_d} - \frac{a_{i+1} - \mu_a}{\rho_a}$ $K = \frac{1}{n}\sum_{i=1}^{n} Ki$ | Dynamic indicator of fluctuation between ambient/focal visual behavior. | Useful to detect too much focusing on a flight instrument (e.g., attentional tunnelling) in a sliding time window. | Z-score comparisons require an exhaustive database of all dwells. Ki close to zero can result from long dwelling periods followed by large saccades or short dwells and small saccades. |
| Sequence analyses | Gaze Transition Entropy (GTE) | $H_t = -\sum_{i=1}^{n} p_i \sum_{j=1}^{n} p_{(i,j)} \log_2 p_{(i,j)}$ | Amount of information needed to describe the dwell transitions. | Useful for comparing visual scanning between subject with a single quantitative value. | The same entropy value may be associated with visual scanning that do not take into account the same AOIs. |
| | Lempel Ziv Complexity (LZC) | $LZC_w = \frac{C_w(n)}{n/\log_2 n}$ | Measure the repetitiveness of sequence by data compression. | Provide a code book with all the patterns encountered and their occurrence, correlate with gaze transition entropy. | Quite similar patterns can be countered as different for instance the sequences 1 - 11 - 1 - 11 and 11 - 1 - 11 - 1 are accounted as two different patterns. |
| | Common N-Gram sequences | $P(w_n \lvert w_{n \, N \lvert 1}^{n1}) = \frac{c(w_{n-N+1}^{n-1}, w_n)}{c(w_{n-N+1}^{n-1})}$ | Number of common sequences in each group. | Useful for comparing visual scanning between group or for comparing within group visual scanning consistency. | Difficulty in highlighting important visual scanning patterns. |

**Fig 1. Overview of the different visual scanning metrics classified by approaches.**

can be extended to three dimensions by considering the location of the previous two dwells, which Norris et al. [41] have described as a second-order Markov chain. Jones et al. [42] showed that transitions matrices are sensitive to flight maneuvers. Based on the transition matrices, Hayashi proposed in 2004 [43] a Hidden Markov Model approach corresponding to different flight tasks. Its works were used afterward to model the dwell patterns of the space shuttle crew [44].

**Transition matrix density.**   Introduced by Goldberg and Kotval [31], the transition matrix density describes the dispersion of attention over time [45]. Transition matrix density provides a single quantitative value by dividing the number of active transition cells (i.e., those containing at least one transition) by the total number of cells. An unusually dense transition matrix (large index value), with most cells filled with at least one transition, can indicate a dispersed, lengthy, and wandering visual scan (this can reflect an extensive search on a display for example) [46]. A sparse matrix can reflect a more efficient and directed search, for example when using a computer software [40], or, in other contexts, can indicate a failure to properly monitor the environment, for example when a novice driver directs his gaze continuously to the road while ignoring/forgetting the rearview mirrors or when a pilot is excessively engaging his visual attention on a single instrument (e.g., [47]).

## Attentional modes

**K coefficient.**   Another evaluation of the dispersion of the attention is a novel parametric scale called K coefficient introduced by Krejtz et al. [48]. This metric was created and developed during exploring artwork (e.g., painting) and map viewing [49] in order to investigate the dynamics of visual scan (focal vs ambient) when operating such tasks. In a recent study, Lounis et al. [50] used this method by modifying input data, using dwells and transitions

instead of fixations and saccades. During various flight phases with automation in a full-flight simulator, they calculated for each pilot the mean difference between standardized values (z-scores) of each transition ($a(i + 1)$) and its preceding $i$th dwell ($di$), where $\mathbf{d_i}$ is the duration of the $i$–$th$ dwell and $\mathbf{a_{i+1}}$ the amplitude of the transition that occurs after the $i$–$th$ dwell. $\mu_d$, $\mu_a$ are the mean dwell durations and transition amplitudes, respectively, and $\rho_d$, $\rho_a$ are standard deviations, respectively.

$$\kappa_i = \frac{\mathbf{d_i} - \mu_d}{\rho_d} - \frac{\mathbf{a_{i+1}} - \mu_a}{\rho_a}, \qquad \kappa = \frac{1}{n}\sum_{i=1}^{n}\kappa_i \tag{1}$$

Values of $K_i$ close to zero indicate relative similarity between dwell durations and transition amplitudes. Positive values of $K_i$ show relatively long dwells followed by short transition amplitudes, which indicate focal attention. Negatives values of $K_i$ refer to the situation where relatively short dwells are followed by a relatively long transition, suggesting ambient attention (diffuse attention). According to Heitz, R. P., & Engle, R. W. (2007) [51], in the diffuse mode, visual attention is more allocated to all regions of the visual field in quite equal proportion; in the focused mode, attention is concentrated at a few areas of interest, specified by a central or peripheral cue. An extremely focused mode could be compared to the concept of attentional tunnelling [47]. It is worth noting that the values of the K coefficient should be interpreted together with dwell duration results because different groups can have different average values of dwell duration and transition amplitudes.

## Sequence analyses

The sequence analyses approach allows measuring the extent to which the time sequence of eye movements is ordered or random during a flight.

**Gaze Transition Entropy (GTE).**  Defined by Shannon and Weaver [52], entropy is a measure of lack of predictability in a sequence. This metric enables evaluating the structuration of the gaze [53]. When applied to eye tracking data, transition entropy describes the amount of information needed to describe the visual strategies, following the formula:

$$\text{GTE} = \sum_{i=1}^{n}p(i)\left[\sum_{j=1}^{n}p\left(\frac{j}{i}\right)\log_2 p\left(\frac{j}{i}\right)\right], \ i \neq j \tag{2}$$

where i represents the "from" AOI and j represents the "to" AOI. Higher transition entropy denotes more randomness and more frequent switching between AOIs [54]. Ephrath, Tole, Stephens, and Young [55] have noticed an increase of entropy with increasing pilots' mental workload (by adding a secondary task). Van de Merwe et al. [56] found that entropy increased as a result of cockpit instrument failure, conditions that most likely produce an increased mental workload. More recently, using GTE, Allsop et Gray, 2014 [57] revealed that visual scanning became more random during the an anxiety landing scenario. Diaz-Piedra et al. [58] observed a significant decrease in pilot's gaze entropy when pilots faced a scenario presenting more complexity.

**Lempel-Ziv complexity.**  The complexity (i.e., the quantity and diversity) of visual scanning patterns can be assessed using Lempel-Ziv Complexity (LZC). LZC was defined by Lempel and Ziv in 1976 (for a review, see [59] as a data compression algorithm computing the minimum number of bits from which a particular message or file can effectively be reconstructed. This algorithm counts the number of different patterns in a sequence when scanned from left to right. For instance, Lempel-Ziv complexity of s = 101001010010111 is 7, because when scanned from left to right, 7 different patterns are observed: 1|0|10|01|010|0101|11.

Recently, LZC was applied to the dwell transition to evaluate the number of different visual scanning patterns [60].

**N-gram sequences.** N-gram is an essential component of many methods in bioinformatics, including for genome and transcriptome assembly, for metagenomic sequencing, and for error correction of sequence reads [61]. Basically, an N-gram model predicts the occurrence of an AOI, based on the occurrence of its N–1 previous AOI. So here we are answering the question: How far back in the history of a sequence of AOI should we go to predict the next AOI? For instance, a bigram model (N = 2) predicts the occurrence of an AOI given only its previous AOI (as N–1 = 1 in this case). Similarly, a trigram model (N = 3) predicts the occurrence of an AOI based on its previous two AOI. The common N-gram sequence analysis used the n-grams frequency-based method [62] to identify the number of common 3, 4, 5, and 6-gram sequences in each group. By using this method, it is possible to count the occurrence of N-gram AOI and their occurrence for each pilots, and thus it allows to compare for each N-gram the intra-group patterns consistency.

## Current study

In the present study, we evaluated the efficiency of the previously describe metrics on the eye tracking data from novice and expert pilots. Our main hypothesizes were that expert pilots should exhibit different visual behaviors than novices, including more numerous dwells and shorter dwell times, following the idea that superior perceptual encoding processing comes with expertise. We expected also a sensitivity of all advanced metrics to expertise, with more visual scanning complexity (as evaluated by the Lempel Ziv complexity and the visual pattern lengths), and a more regular visual scanning (as evaluated by the transition entropy) in experts. We also assumed that the pilots' expertise could be classified (using machine learning) in their way that they switched from an instrument to another, using transition matrices. Finally, we hypothesized that the addition of a parallel monitoring task should also have an impact on ocular behavior, notably by increasing complexity, reducing the regularity level, and generating an ambient mode of attention (i.e. more diffuse attention).

## Materials and methods

For reproducibility purpose, the protocol is available on protocols.io; DOI number: dx.doi.org/10.17504/protocols.io.zb5f2q6.

## Participants

Thirty-two participants, all males, participated in this experiment. They all had normal or corrected to normal vision. They were not informed about the exact purpose of the study. They were divided into two groups according to their flying experience. A first group called "novices" consisted of participants with no real flight experience (n = 16, mean age 25.7±5.5 years). They were recruited from a French aerospace engineering school (ISAE-SUPAERO, Toulouse, France). All these novices participants had advanced theoretical knowledge about aeronautical engineering, were familiar with the various information given by the instruments in the cockpit (altimeter, altitude etc.), and had flight notions on how to manually interact with the aircraft. Our experimental flight scenarios were relatively simple: the participant had to control the trajectory and the speed of the aircraft. The scenarios did not require complex navigation activities or interacting with automation. Thus the scenarios were feasible for these novices after a relatively short training session. A second group called "pilots" consisted of active professional airline pilots (n = 16, mean age 34.39 ± 8.86 years) with a minimum of 1600 flight hours (mean = 4321.73 ± 2911.41 hours). They were recruited from various airline companies.

They all flew on A320 and were currently flying on A320 (68.75%) or B737 (31.25%) at the time of the experiment.

## Ethics statement

This research project was approved by the local institutional Research Ethics Committee of the University of Toulouse (Comité d'Ethique de la Recherche de l'Université de Toulouse, code N˚2019-131) and was conducted in accordance with the Helsinki Declaration. Volunteers signed an informed consent prior to the experiment and were informed of their right to stop their participation at any time.

## Materials

**Flight simulator.** We used an A320-like flight simulator ("PEGASE") located at ISAE-SU-PAERO (Toulouse, France), see Fig 2. Like in the A320 aircraft, flight instruments included a Primary Flight Display (PFD), a Navigation Display (ND), an Electronic Central Aircraft Monitoring display (ECAM), and an FCU (Flight Control Unit). The field of view covered by the simulator is about 180˚. The participants controlled the aircraft with a side-stick, two thrust

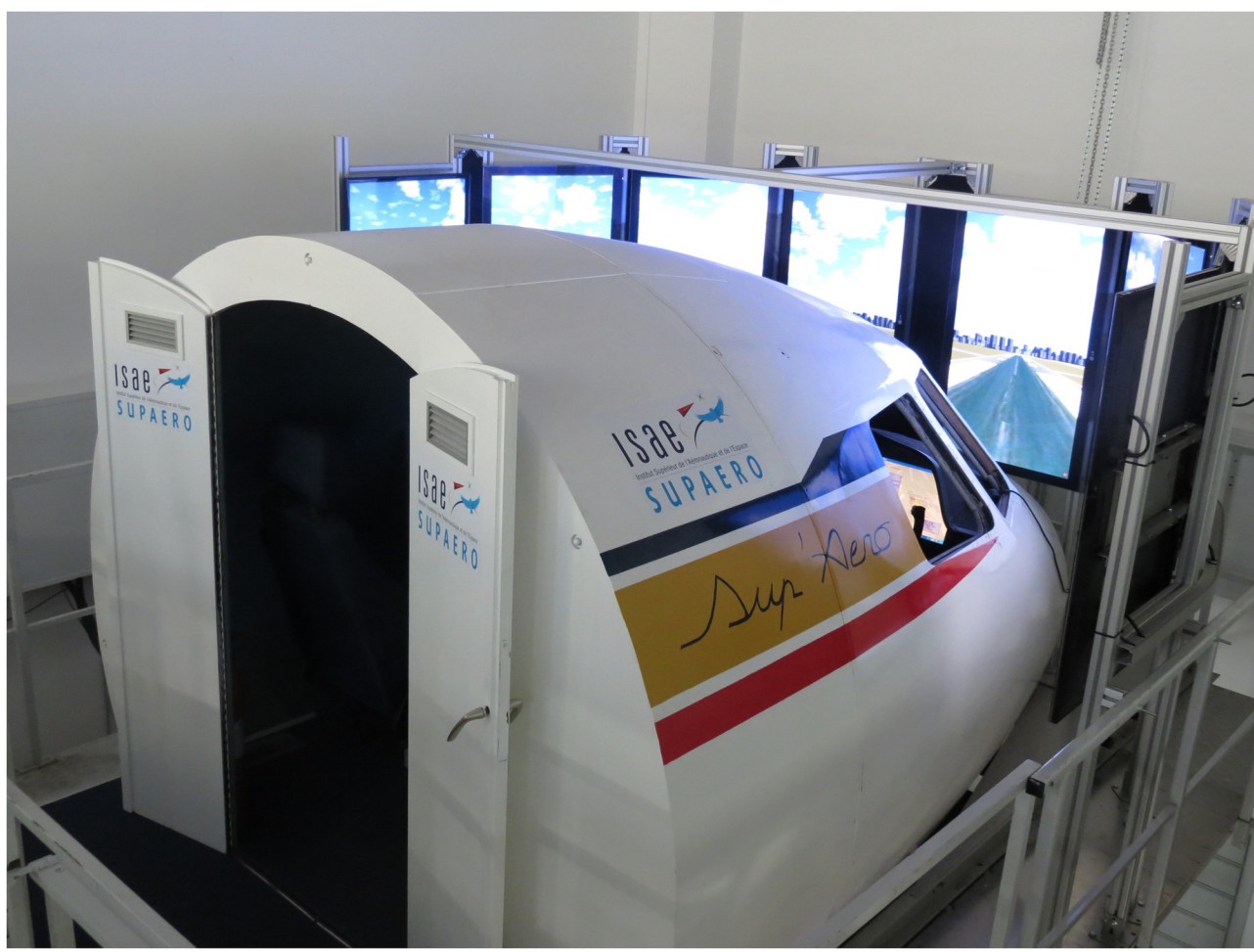

**Fig 2. ISAE-SUPAERO flight simulator with its external screens.**

levers, and a rudder. We recorded flight data to calculate flight performance during the landing.

**Flight scenarios.** Participants manually (i.e., without the autopilot) performed three times the same landing scenario according to three different conditions. The "control scenario" (CS) was a nominal landing without a supplementary task. The "easy dual task scenario" (EDTS) and the "difficult dual task scenario" (HDTS) were similar to the "control scenario" except that participants were asked to perform a supplementary monitoring task. The purpose of this supplementary task was to increase the level of visuo-attentional effort: participants had to regularly check the ND Zone in the ND screen to say aloud the value at the right time. In the "easy dual task scenario", participants were asked to say aloud the distance between the aircraft and the airfield threshold every 0.5 Nm (information provided by a radio beacon localized near the airfield and displayed in the ND Zone, see Fig 3). In the "difficult dual task scenario", they were asked to say aloud this distance every 0.2 Nm. The experimenter stayed in the cockpit during the entire experimentation. Each of the three-landing scenarios consisted of performing an approach/landing to Toulouse-Blagnac Airport, Runway LFBO 14R. The flight began at coordinates 1.2159˚ of longitude and 43.7626˚ of latitude. During each scenario, the participants had to comply with the same specific instructions related to the flight. In particular: to maintain a vertical speed between +500 ft/min and -800 ft/min, a speed of 130 knots, and a heading of 143˚ (corresponding to the Runway 14R). We choose these values because they roughly correspond to a standard landing speed with a commercial aircraft. The negative vertical speed of -800 ft/min approximately corresponds to the vertical speed at 130 kt with an angle of approach of three degrees. We defined a tolerance range in case the participant was not well stabilized on the approach slope and had to regain altitude (+500 ft/min maximum). Each landing scenario started at an altitude of 2000 ft and lasted approximately four minutes. The three scenarios were randomized across participants to avoid learning effects. Performance dependent variables were heading, vertical speed, and speed deviations. The number of omissions (i.e., the participant omitted to call out the distance) during the supplementary task was also calculated.

**Eye movements recordings.** Eye movements were recorded at 60Hz using a Smart Eye remote eye tracker (Smart Eye AB, Sweden). The system detects human face/head movements, eye movements, and gaze direction. Gaze direction and eyelid positions are determined by

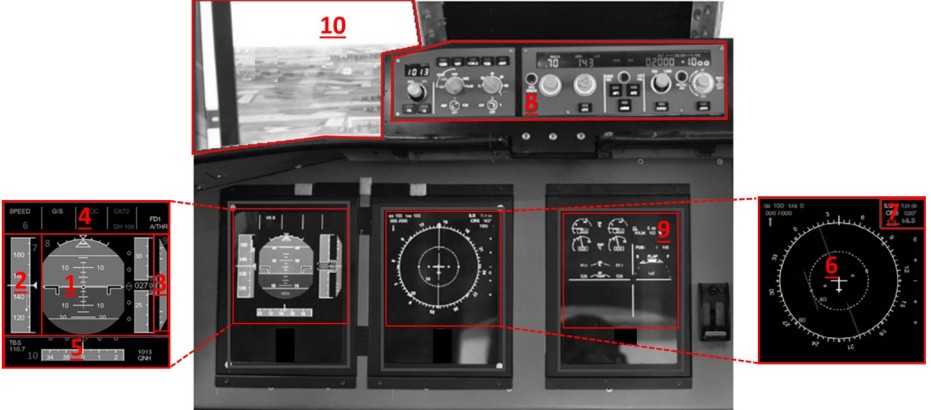

**Fig 3.** Overview of the ten different AOIs: (1) Attitude indicator, (2) Speed tape, (3) Vertical speed tape, (4) Flight mode annunciator, (5) Heading tape, (6) Navigation display, (7) ND zone (displays the distance to recall during the two landing scenarios with the supplementary task), (8) Flight control unit, (9) Electronic centralized aircraft monitoring, (10) Out of the window.

combining image edge information with 3-D models of the eye and eyelids. As presented in Fig 3, the system uses five cameras integrated into the cockpit. A major advantage of using several cameras is that eye and head tracking can be maintained despite significant head motions (translation and rotation) or occlusion of one of the cameras by the participant (e.g., by its hand). Smart eye system allows to design a 3D environment and to establish calibration points (at the vicinity of AOI). When the world model is designed, we just need to operate an automatic calibration for each participant.

**World model and area of interest.** The cockpit was split into 10 AOIs, corresponding to the different flight instruments and displays that pilots can examine during a flight, see Fig 3. We choose to restrict our analysis to instruments that display information directly related to the flight parameter (altitude, speed etc.) and external view (i.e., Out of the Window).

## Procedure

At first, participants filled out the consent form and provided demographic information such as their flight qualification (aircraft type) and their flight experience (total hours of flight experience). Participants were briefed on the study and instructed about the different flight scenarios. Then, they were invited to seat in the flight deck at the captain position (left seat). The eye-tracking system was calibrated using an 11-point calibration. Following the Smart Eye manual recommendation, the 11 points were located in the vicinity of the AOIs. Participants performed a training session, consisting of performing two times a landing scenario. All participants (including novice ones) were able to control the aircraft correctly after these two landing. Then, the participants performed three times the same landings scenario than during the training, but with varying levels of complexity.

## Data processing

Flight simulator and eye-tracking data were analyzed using MATLAB R2019b with custom homebuilt scripts. The data were recorded from the beginning of the landing scenario to touch-down. Because the landing duration depends on the pilot's actions, landing durations could differ by a few seconds. As a consequence, the beginning of the scenarios has been cut out to obtain the same duration for each participant, corresponding to 14,000 frames sampled at 60 Hz for the eye-tracking data and 233 frames at 1 Hz for the flight simulator.

**Eye tracking data.** Fig 4 shows the entire eye tracking pipeline analysis. Each AOI was coded using numbers from 1 to 10 corresponding to the flight instruments (see Fig 3). Only AOI-based data were extracted in this experiment and concatenated to obtain two chronological vectors containing the indices of the visited AOIs (from 1 to 10) and the time spent on them. Dwells inferior to 200 ms [40] were discarded. Furthermore, consecutive fixations in the same area were merged (e.g., for 1, 1, 4, 4, 5, 5, 5, 6 we only consider 1, 4, 5, 6). The transition vector (the vector containing the transitions between each AOI numbers) was used to compute LZC, GTE. Concerning the transition matrices, given their high dimensionality, it is difficult to use classical inferential statistics. Therefore, we applied machine learning algorithms on the concatenated transition matrices to compare the two groups of participants (novice vs pilot). Various types of machine learning model were used (SVM, LDA, K-Nearest Neighbor, for a review see [63]). The algorithm performing the best accuracy (Cosine KNN) was selected in this paper. The transition probabilities from one AOI to another were taken as a feature, thus raising the number of features to a total of 100 features (i.e., 10 AOIs × 10 AOIs). A principal component analysis (PCA) was used to reduce the features' numbers. This restricts the model to 35 features corresponding to the main transition probabilities of the matrices. Five-fold cross-validation was used, which is a good trade-off between bias and variance estimation [64].

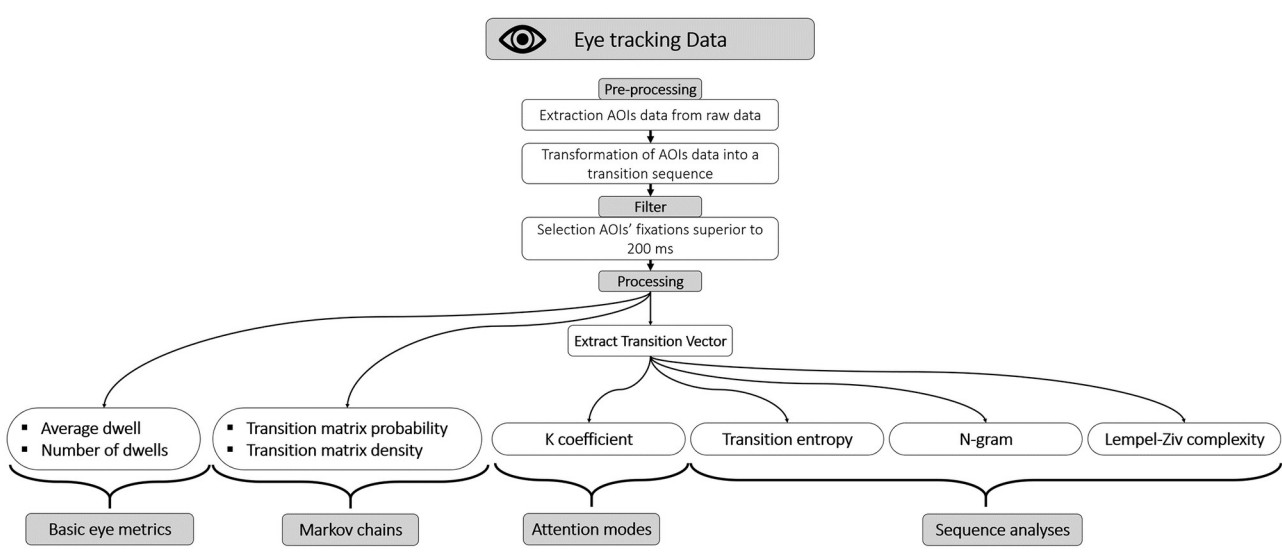

**Fig 4. Analysis pipeline for the eye tracking data.**

According to Combrisson and Jerbi [65] theoretical chance level for classification for $p < 0.05$ with two classes is around 58%. Concerning the K coefficient, the transition entropy, and the Lempel Ziv complexity methods, they were respectively computed following the methods of [48, 54, 60]. Finally, based on the transition vector, the n-grams frequency-based method [62] was used to identify the number of common 3, 4, 5, and 6-gram sequences in each group. After counting the occurrence of given n-grams for each participant, the number of common sequences of each n-gram was calculated for each group (Novice/Pilots).

**Flight simulator data.** The flying performances were examined to quantify the ability of the pilot to comply with the specific flying instructions given by the experimenter. As presented in Fig 5, Root Mean Square Errors (RMSEs) were calculated for 3 different flight parameters: speed, vertical speed, and heading. In this experiment, the predicted values corresponded to the different specific threshold given by the experimenter (i.e., speed 130 kt; vertical speed below -500 ft/min and above +800 ft/min; heading different from 143˚) and the observed values corresponded to actual pilots' performances. The deviations were calculated following the formula:

$$\text{RMSE}_{k,k+1} = \sqrt{\frac{1}{n}\sum_{i=1}^{n}(O_i - P_i)^2} \tag{3}$$

as where for n data points between points $k$ and $k + 1$, $Pi$ was the predicted value and $Oi$ the observed value.

## Statistical analysis

We performed $2 \times 3$ repeated measures analysis of variance (ANOVA) for each dependent variable (i.e., dual task omission, average dwell time, the total number of dwells, LZC, transition entropy, K coefficient, RMSE heading, RMSE vertical speed, RMSE speed) to assess the effects of the group (novices, pilots) with scenario difficulty as the within-subjects factors (three levels: Control scenario, Easy dual task scenario, Difficult dual task scenario). The normal distribution for each dependent variable was also checked. We used the Greenhouse-Geisser and

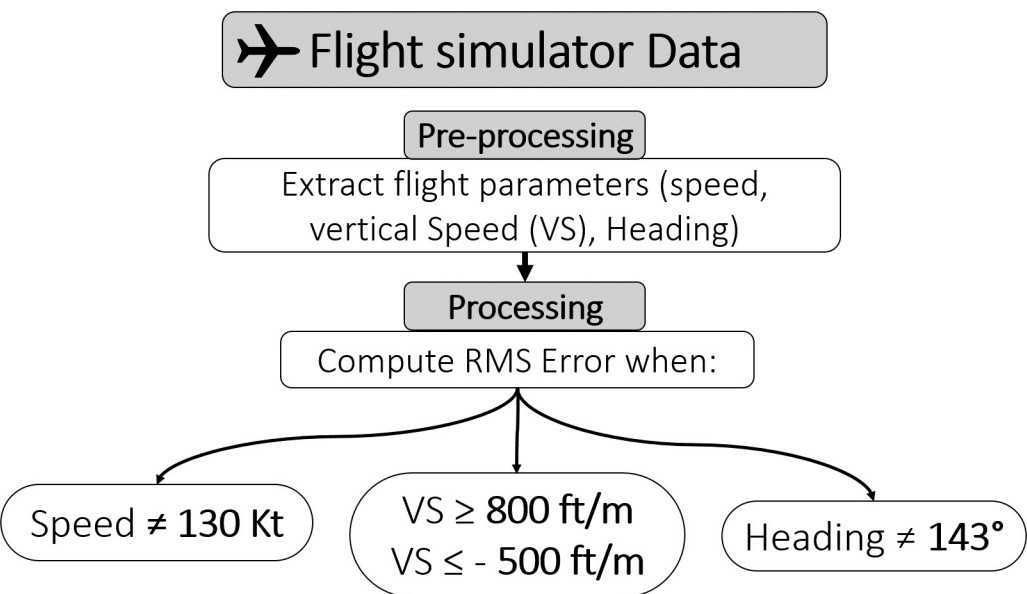

**Fig 5. Analysis pipeline for the flight parameters data.**

Huynh-Feldt adjustment to correct the violation of the sphericity assumption when needed. Bonferroni post-hoc tests were performed for multiple comparisons and reported Bonferroni post-hoc are only those with significant differences. The level of significance was set to $\alpha = 0.05$ and partial $\eta^2$ was used to estimate the effect sizes.

## Results

### Flight performances

The flight performances are shown in Fig 6.

**Heading.** There was no significant main effect of the group, $F(1, 30) = 0.03$, $p = 0.874$, nor main effect of the scenario, $F(2, 60) = 0.9$, $p = 0.39$, on heading deviations. The scenario × group interaction was not significant, $F(2, 60) = 0.4$, $p = 0.67$.

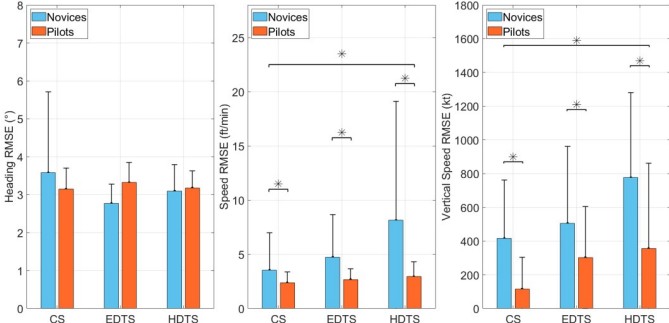

**Fig 6. Flight performances for heading, vertical speed, and speed deviations among novices and pilots groups.**
Error bars represent SD and * indicates main effects $p < 0.05$. (CS = control scenario; EDTS = Easy dual task scenario; HDTS = Hard dual task scenario).

**Speed.** A significant main effect of the group on speed deviation was found, $F(1, 30) = 4.3$, $p < 0.05$, $\eta^2 = 0.13$, with the novice's group (M = 5.46; SD = 1.94) showing higher speed deviation than pilot's group (M = 2.66; SD = 1.97). Analyses also revealed a significant main effect of the scenario, $F(2, 60) = 3.6$, $p < 0.05$, $\eta^2 = 0.11$. Bonferroni post-hoc test showed that speed deviation was lower during the control scenario (M = 2.95; SD = 0.93) compared to the easy dual task scenario (M = 3.70; SD = 1.02) and the difficult dual task scenario (M = 5.53; SD = 2.79). There was a significant effect of scenario × group interaction, $F(2, 60) = 3.3$, $p < 0.05$, $\eta^2 = 0.09$. Bonferroni post-hoc test showed that the speed deviation was lower for the pilot's group in the difficult dual task scenario (M = 2.93; SD = 3.97) compared to the novice's group in the difficult dual task scenario (M = 8.13; SD = 4.02).

**Vertical speed.** Analyses revealed a significant main effect of the group, $F(1, 30) = 11.4$, $p < 0.05$, $\eta^2 = 0.28$, on vertical speed deviation, with the novice's group (M = 565; SD = 130) showing higher vertical speed deviation than pilot's group (M = 258; SD = 134). Analyses also revealed a significant main effect of the scenario, $F(2, 60) = 5.1$, $p < 0.01$, $\eta^2 = 0.15$. Bonferroni post-hoc test showed that the vertical speed deviation was lower during the control scenario (M = 265; SD = 103) compared to the easy dual task scenario (M = 403; SD = 141) and the difficult dual task scenario (M = 566; SD = 184). The scenario × group interaction was not significant, $F(2, 60) = 0.7$, $p = 0.52$, $\eta^2 = 0.02$.

## Dual task omissions

Analyses showed (Fig 7) a significant main effect of the group on omissions, $F(1, 30) = 35.3$, $p < 0.05$, $\eta^2 = 0.54$. The novice's group had a higher number of omissions (M = 2.75; SD = 1) than the pilot's group (M = 0.68; SD = 0.5). Analyses also revealed a significant main effect of the scenario, $F(1, 30) = 24.8$, $p < 0.05$, $\eta^2 = 0.45$. Bonferroni post-hoc test showed that the difficult dual task scenario (M = 2.37; SD = 0.52) yielded more omissions than the easy dual task scenario (M = 1.06; SD = 0.3). The scenario × group interaction was significant, $F(1, 30) = 16.2$, $p < 0.05$, $\eta^2 = 0.35$. Bonferroni post-hoc test showed that there were more omissions during the difficult dual task scenario (M = 1.5; SD = 1) vs. easy dual task scenario (M = 3.95; SD = 2) in novices whereas the number of errors did not differ among the two scenarios for pilots.

## Basic eye metrics

**Average dwell times.** Analyses showed (Fig 8) a significant main effect of group, $F(1, 30) = 8.1$, $p < 0.05$, $\eta^2 = 0.22$, with short average dwell times for the pilot's group (M = 1.1;

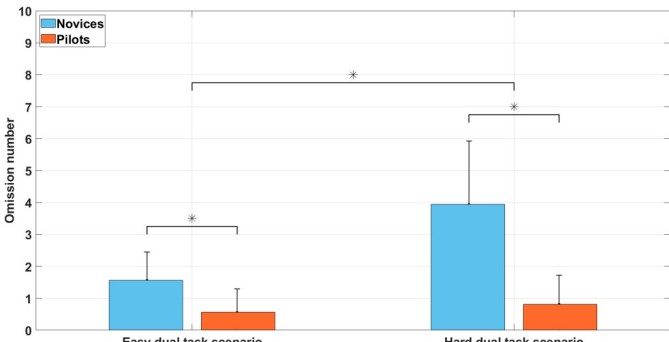

**Fig 7. Omission number for the easy dual task scenario and hard dual task scenario among novices and pilots groups.** Error bars represent SD and * indicates main effects $p < 0.05$.

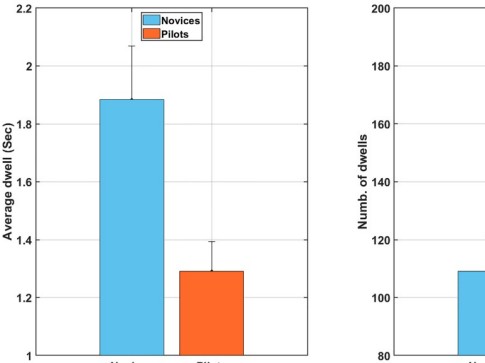

**Fig 8. From left to right, respectively the average dwell and the number of dwells averaged over all scenarios among novice and pilot groups.** Error bars represent SD and * indicates main effects $p < 0.05$.

SD = 0.2) compared to the novice's group (M = 1.51; SD = 0.21). We also found a significant main effect of the scenario, $F(2, 60) = 19.0$, $p < 0.05$, $\eta^2 = 0.39$. Bonferroni post-hoc showed that the average dwell time was shorter during easy dual task (M = 1.16; SD = 0.12) and difficult dual task scenario (M = 1.16; SD = 0.17) than during the control scenario (M = 1.58; SD = 0.22). There was no significant scenario × group interaction, $F(2, 60) = 2.3$, $p = 0.11$, $\eta^2 = 0.07$. The time spent gazing outside the defined AOIs was relatively low (M = 4.21% for experts; M = 4.62% for novices), see supplementary material for detailed information (S1 Fig).

**Number of dwells.** Analyses showed (Fig 8) a significant main effect of group, $F(1, 30) = 13.3$, $p < 0.05$, $\eta^2 = 0.31$, with a higher number of dwells for the pilot's group (M = 188; SD = 21) compared to the novice's group (M = 137.5; SD = 19.9). Analyses also revealed a significant main effect of the scenario, $F(2, 60) = 13.2$, $p < 0.05$, $\eta^2 = 0.31$. Bonferroni post-hoc showed that the number of dwells was higher during easy dual task scenario (M = 172; SD = 16) and during the difficult dual task scenario (M = 177; SD = 18) compared to the control scenario (M = 137; SD = 17). There was no significant scenario × group interaction, $F(2, 60) = 0.7$, $p = 0.50$, $\eta^2 = 0.02$.

## Markov chain and machine learning

The confusion matrix presented in Fig 9 show that approach based on Cosine KNN reached classification accuracy up to 91.7% to classify expertise based on transition matrices during the baseline scenario. As shown in Fig 10, the differences in transition matrices between novices/pilots are mainly observed in a more sparsed distribution of transition probabilities from one instrument to another for the pilot's group. Most of the AOI explored by Novice group involved AOI concentrated in the PFD (from 1 to 5, see Fig 4) while pilot's group explore other combinations of AOI.

## Attentional modes and K coefficient

Analyses showed (Fig 11) no significant effect of the group, $F(1, 30) = 3.3$, $p = 0.07$, $\eta^2 = 0.10$, on the $K$ coefficient. However, the main effect of scenario was significant, $F(2, 60) = 38.1$, $p < 0.01$, $\eta^2 = 0.56$. Bonferroni post-hoc test showed that K coefficient was lower during the easy dual task scenario (M = -0.12; SD = 0.06) and during the difficult dual task scenario (M = -0.01; SD = 0.12) compared to the control scenario (M = 0.28; SD = 0.10). There was also a significant difference between the easy dual task scenario (M = -0.12; SD = 0.06) and the difficult dual task scenario (M = 0; SD = 0.12). The scenario × group interaction was significant, F(2,

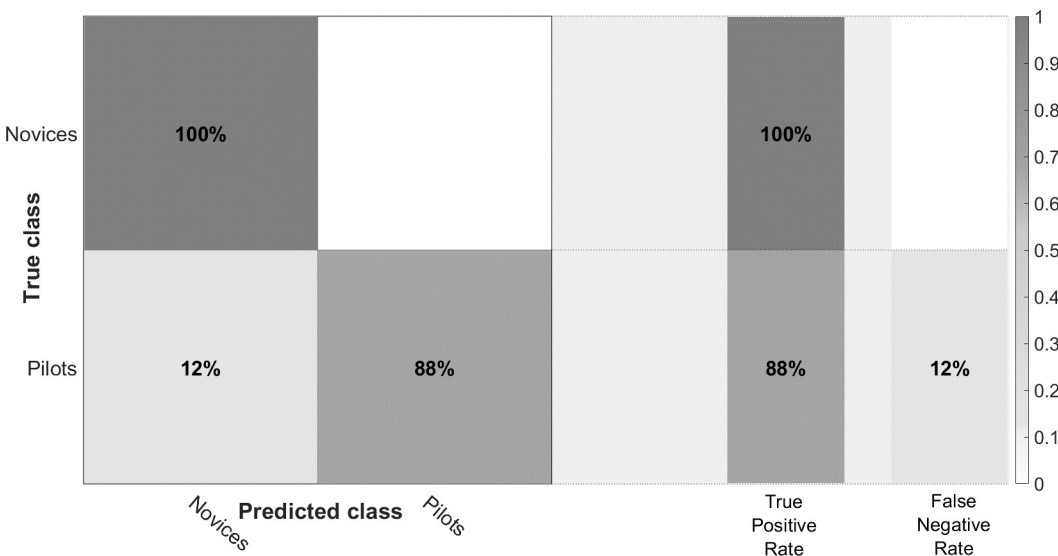

**Fig 9. Confusion matrix of fivefold cross-validation using the Cosine K-Nearest neighbors among novices and Pilots groups during the baseline scenario.**

60) = 4.8, $p$ = 0.01, $\eta^2$ = 0.15. Bonferroni post-hoc test showed that K coefficient was lower for the pilot's group in the control scenario (M = 0.14; SD = 0.16) compared to the novice's group in the control scenario (M = 0.41; SD = 0.16). Bonferroni post-hoc test also showed that K coefficient was lower for the pilot's group in the difficult dual task scenario (M = -0.10;

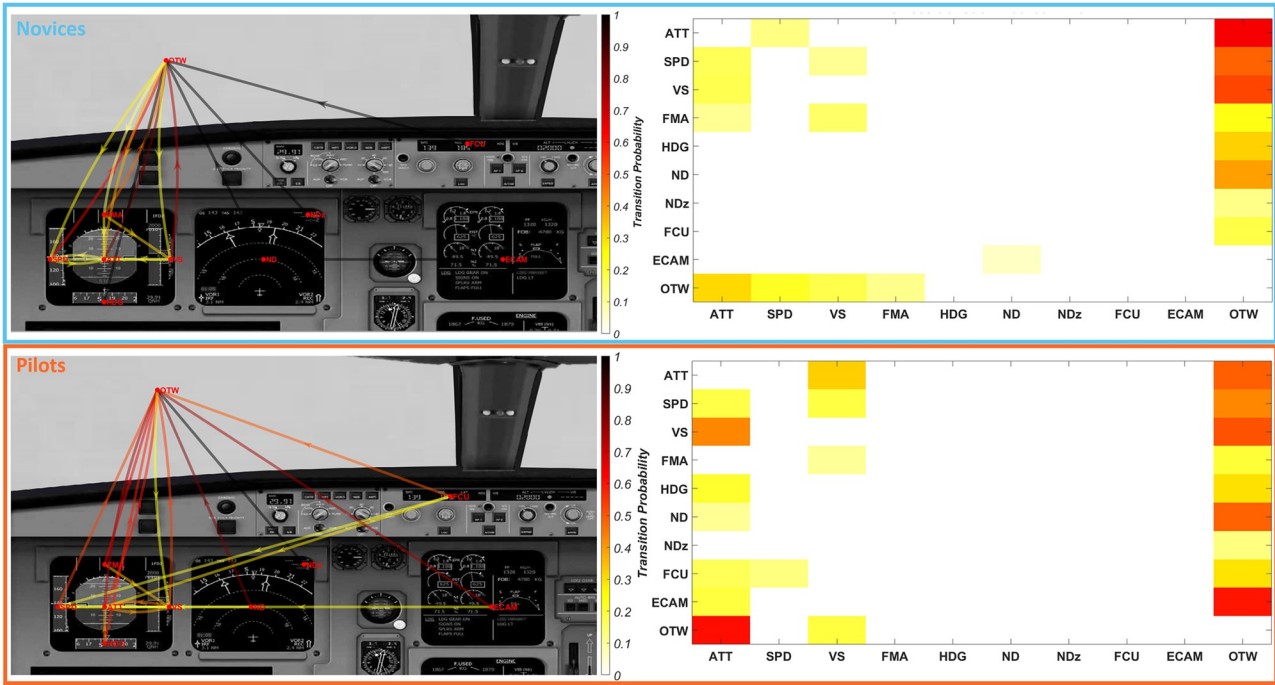

**Fig 10. Markov chains (Left) and transition matrices (Right) AOI-based representations among novices (top) and pilots groups (bottom) during the baseline scenario.**

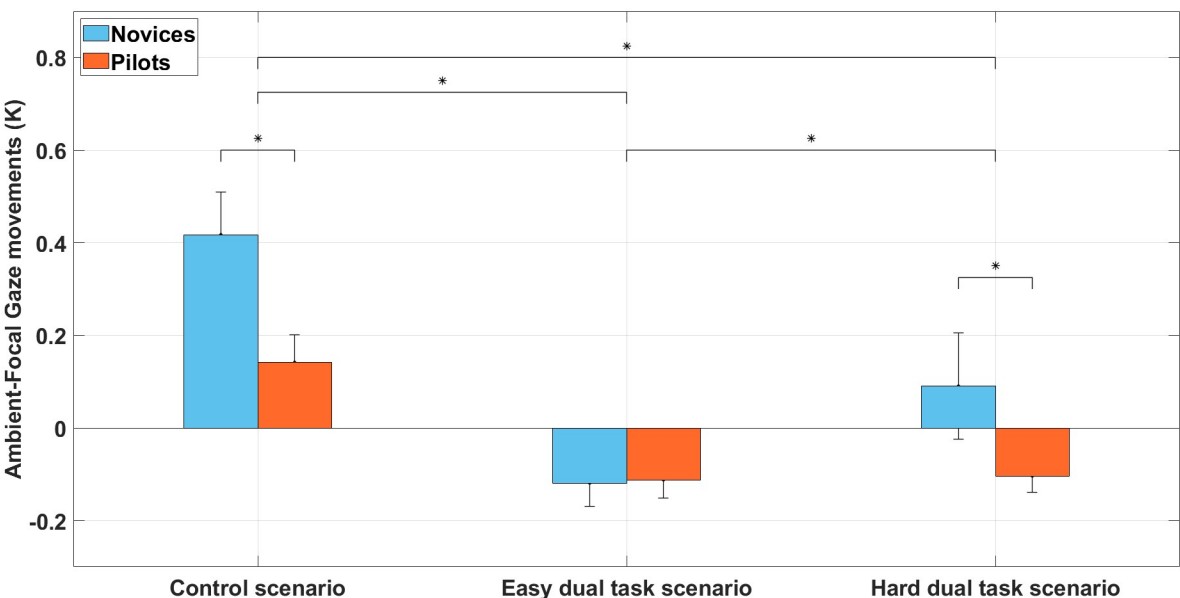

**Fig 11. Ambient focal K coefficient during the control scenario, the easy dual task scenario, and hard dual task scenario among novices and pilots groups.** $k > 0$ indicates a focal visual attention, whereas $k < 0$ indicates an ambient visual attention. (error bars represent SD and $^*$ indicates main effects $p < 0.05$.

SD = 0.17) compared to the novice's group in the difficult dual task scenario (M = 0.09; SD = 0.16).

## Sequence analyses

**Transition entropy.** Analyses showed (Fig 12) a significant main effect of group, $F_{(1, 30)} = 6.0$, $p < 0.05$, $\eta^2 = 0.17$, with the novice's group (M = 1.22; SD = 0.2) showing lower transition entropy than pilot's group (M = 1.56; SD = 0.2). Analyses also revealed a significant main effect of the scenario, $F_{(2, 60)} = 8.4$, $p < 0.05$, $\eta^2 = 0.22$. Bonferroni post-hoc test showed that the transition entropy was higher during easy dual task scenario (M = 1.50; SD = 0.16) and during difficult dual task scenario (M = 1.44; SD = 0.17) than during the control scenario

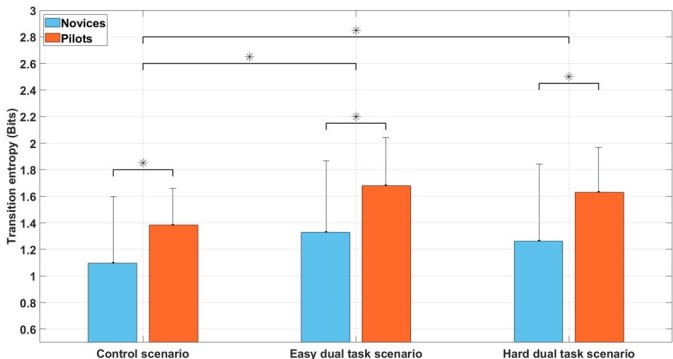

**Fig 12. Transition entropy during the control scenario, the easy dual task scenario, and the hard dual task scenario among novices and pilots groups.** Error bars represent SD and $^*$ indicates main effects $p < 0.05$.

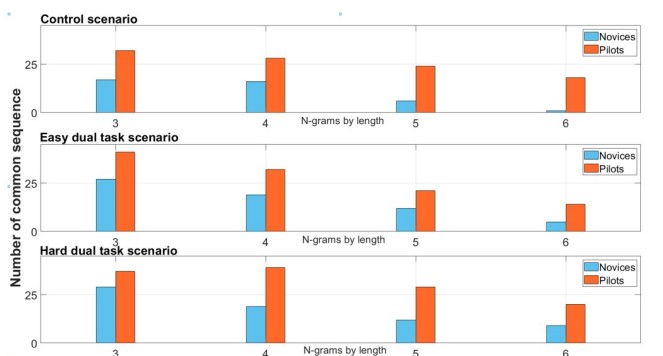

**Fig 13. Number of common patterns sequence by N-grams length during the control scenario, the easy dual task scenario, and the hard dual task scenario among novices and pilots groups.**

(M = 1.23; SD = 0.15). The scenario x group interaction term was not significant, F(2, 60) = 0.2, $p$ = 0.82, $\eta^2$ = 0.01.

**Common N-grams sequence.** As presented in Fig 13, the count of common n-gram sequences revealed that pilots have more common sequences than novices during all scenarios (Control, easy dual task, and hard dual task). The easy dual task and hard dual task scenario yielded to more common sequences for both groups compared to the control scenario. Regardless of the n-gram length (3, 4, 5, or 6), during the control scenario, the pilots had more common sequences than novices. For example, the most frequent tri-gram pattern for the novices was OTW⇒VS⇒OTW—transition between out-of-the-window, vertical speed, and back. On average, it was repeated 6.4 times. For the pilots, the most frequent tri-gram occurred 17.4 times on average and it was OTW⇒ECAM⇒OTW. We also note that the ten most frequent n-grams included the same AOI at least twice (for instance, repeated transitions between same instruments). For novices, trigrams involving three unique AOIs were OTW⇒SPD⇒ATT repeated 2.6 times on average, OTW⇒VS⇒ATT—2.1 times, and OTW⇒ATT⇒SPD—2 times. For pilots, the trigrams involving unique AOIs were OTW⇒ECAM⇒ATT repeated 9.4 times on average, OTW⇒VS⇒ATT—8.6 times, OTW⇒ATT⇒VS—4.6 times, and OTW⇒HDG⇒ATT—3.8 times. For the both easy and hard dual task scenarios, the most frequent trigram involved the ND zone display for both groups OTW⇒NDz⇒OTW. It occurred 17.6 times on average for novices and 19.1 for pilots during the easy dual-task scenario, and 21.1 times on average for novices and 22.2 for pilots during the hard dual-task scenario. Our results showed that for novices only one frequent trigram with unique AOIs found in the control scenario was also found during the easy dual task scenario (OTW⇒VS⇒ATT). However, this trigram was not found during the hard-dual task scenario. As for the pilots, between four trigrams with unique AOIs that were found in the control scenario, only 2 of them were found in the easy dual-task scenario, and only one in the hard dual task scenario (see Table 1). Interestingly, the most frequent 5-grams among novices was OTW⇒SPD⇒OTW⇒SPD⇒OTW repeated on average 1.5 times whereas OTW⇒VS⇒ATT⇒OTW⇒ATT was the most frequent 5-gram among pilots repeated on average 3 times.

**Lempel-Ziv Complexity (LZC).** Analyses showed (Fig 14) a significant main effect of group, F(1, 30) = 10.0, $p < 0.05$, $\eta^2$ = 0.25, with a higher LZC for the pilot's group (M = 40.3; SD = 5.6) compared to the novice's group (M = 33; SD = 5.2). There was also a significant main effect of the scenario, F(2, 60) = 13.2, $p < 0.05$, $\eta^2$ = 0.30. Bonferroni post-hoc test showed that LZC was higher during easy dual task (M = 40.46; SD = 4.4) and difficult dual task

**Table 1. The most frequent trigrams involving unique AOIs in the pilot group during the Control Scenario (CS), the Easy Dual-Task Scenario (EDTS), and the Hard Dual-Task Scenario (HDTS).**

| Frequent trigram with unique AOIs | Av. occur. in the CS | Av. occur. in the EDTS | Av. occur. in the HDTS |
|---|---|---|---|
| OTW⇒ECAM⇒ATT | 9.4 | 0 | 0 |
| OTW⇒VS⇒ATT | 8.6 | 7.7 | 0 |
| OTW⇒ATT⇒VS | 4.6 | 5.5 | 0 |
| OTW⇒HDG⇒ATT | 3.8 | 0 | 0 |

Av. occur. = average occurences.

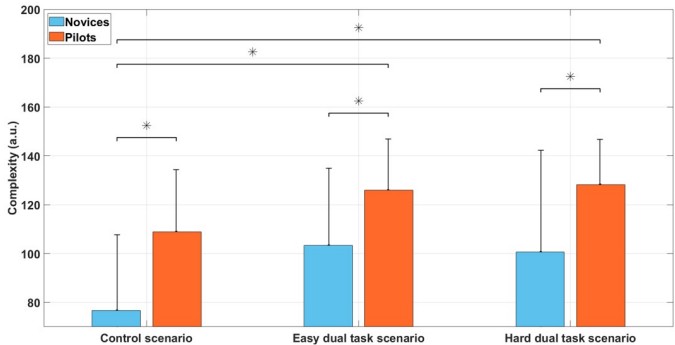

**Fig 14. Lempel-Ziv complexity during the control scenario, the easy dual task scenario, and hard dual task scenario among novices and pilots groups.**

scenario (M = 37.9; SD = 4.97) than during the control scenario (M = 31.7; SD = 3.76). The scenario × group interaction was not significant, F(2, 60) = 0.5, $p$ = 0.62, $\eta^2$ = 0.02.

## Discussion

Several previous studies have reported differences among pilots and novices in how they scan cockpit instruments using standard metrics such as fixation duration, dwell times, numbers of saccades, etc. In this work, standard and advanced eye metrics were analyzed in sixteen novices and sixteen professional pilots during landing scenarios involving different visuo-attentional effort. All the metrics used in this study allowed characterizing visual scanning. We examined the impact of expertise and flying difficulty on the visual scanning strategies. As our results showed, a large number of standard and advanced metrics were sensitive to these two factors. Each metric has its strengths and weaknesses to bring an understanding of visual strategies. For instance, while a transition matrix measure and an entropy value are closely related, the information presented for one and the other is different. A transition matrix makes it possible to measure the preferred paths when consulting AOIs. It highlights the strength of the links between AOIs while the entropy will reflect the disorder of these transition sequences. The application of these metrics can be different. For example, if the aim is to redesign a cockpit panel, transition matrices can be very useful because they give the strength of the relationship between AOIs. This metric can allow to bring close together instruments that are often gazed consecutively, which would help to spare the pilot's visual attention effort. Concerning LZC and N-gram method, N-gram compares the patterns used within the group, while LZC assesses the compressibility of the patterns (how varied the patterns are).

### Flight performances

Our results showed that expert pilots had better flying performance than novices. In particular, they had lower speed and vertical speed deviations. The heading variable was not sensitive most likely because the aircraft was nearly in front of the airfield at the beginning of the scenarios. We assume that these superior flying performances are at least partially due to better visual scanning strategies gained with expertise.

### Basic eye metrics

Expert pilots had shorter average dwell times and a higher number of dwells compared to novices. This result has been interpreted in the literature [15, 66, 67] as an important sign of expertise, built on an optimization of the visual information processing, allowing faster extraction of information when consulting a flight instrument. This strategy allows consulting more often the various instruments, resulting in a better updating of situation awareness [68]. This result also validates the existence of a superior perceptual encoding of domain-related patterns [40] in expert pilots.

### Markov chains and attentional mode

Based on transition matrices, a machine learning approach using Cosine KNN algorithm reached an accuracy of 93% to classify expertise. Expert pilots had more heterogeneous transition probabilities when switching from an instrument to another. This suggests that experts include more flight instruments in their visual scans and succeed to balance their time between them. The focal-ambient K coefficient showed that attention was dominantly focal (positive value) in both groups during the control scenario. However, the attention was more focal in the novice's group vs the pilot group. It can be assumed that expert pilots have a greater spatial distribution of their visual attention than novices. The K coefficient also showed sensitivity to the task difficulty. By adding a monitoring task (Easy dual-task scenario) inducing a supplementary display to monitor, visual attention switched from focal to ambient for the 2 groups. Interestingly, by further increasing the time pressure of the monitoring task (hard dual-task scenario), we found that while the induced dual-task changed the ambient-focal strategy of the novices, the pilot group kept their strategy consistent across experimental scenarios.

### Sequence analyses

As showed by the transition entropy analysis, more information (bits) was required to describe expert pilots' visual strategies than the novice group. Thus, the pilot group exhibited more complex visual scanning patterns. The n-grams analysis of common sequences highlighted the existence of more similar visual strategies (within the professional pilot group) built with expertise as well as more elaborate visual strategies considering common visual scanning patterns of size 6 (6-grams). Furthermore, this analysis revealed that some complex patterns (that include only distinct flight instruments) found in the control scenario were still present in both easy and hard-dual task scenarios. We expected that adding a double task would impact the visual scanning. Our results revealed that pilots kept their visual scanning strategies related to the manual landing task by maintaining visual patterns (found in the control scenario) in the dual-task scenarios (easy and hard). We back these results up with the dual task performances and flight performances where maintaining patterns related to the landing task (control scenario) during dual task scenarios (easy dual task and hard dual task) would maintain relevant visual activity for maintaining flight performance and performing callbacks. Finally, AOI redundancies were also found in both groups, i.e. n-grams having twice several same AOI

in an n-gram sequence. The complexity of the Lempel-Ziv demonstrated that redundancies were lower in the pilots group. Pilots displayed a higher complexity and richness of visual patterns, containing a larger variety of possible combinations.

## Expertise theories

Three theories can explain the expert superiority in visual domains. First, the theory of long-term working memory [69] assumes that expertise extends the capacities for information processing. This theory assumes that the limited-capacity assumption should be reconsidered when related to an expert's specific domain. Related to this hypothesis, experts encode and retrieve information more rapidly than novices. This expert's rapid information processing is reflected in shorter dwell durations. The second theory is related to the information-reduction hypothesis [70]. This assumes that expertise optimizes the amount of processed information by neglecting task-irrelevant information. Our results demonstrated that expert' group keeps maintaining the visual scanning strategies related to the piloting activity during the hard dual task scenario while novice under-performed during this scenario. This result highlights the expert's ability to focus toward relevant information to perform the task neglecting redundant information. Eventually, the third theory is the holistic model of image perception [71]. It focuses on the extension of the visual span. Charness et al. [72] shown that experts extract information from widely distanced and parafoveal regions, producing patterns of saccadic selectivity by piece saliency [73]. Our results suggested that expert over-performed the novice group in maintaining their speed. N-gram analysis revealed the visual scanning strategies related to speed were not found for the pilots whereas novices presents this AOI in their sequences. These results suggested the ability for experts to process information through parafoveal processing.

## Limitations

There are some limitations to this study. We compared professional pilots with non-pilots only. The comparison of these two very different profiles can artificially increase the observed differences in terms of ocular behavior. A further research should consider participants with different levels of expertise from novice to expert (e.g., every 1000 hours) to finely examine the implementation of the visual strategies with expertise. Another limitation concerns the flight simulator used in the study. While it is somehow representative and allows to simulate real flight with all primary displays, one should consider a full flight simulator to better fit with the operational context. This experiment could be also replicated with different meteorological conditions. Finally, the eye tracker devices are more and more mature and accurate (about $1˚$ at a distance of one meter). However, care should be taken when analyzing contiguous AOIs, the accuracy limitations of eye tracking systems could lead to errors in this situation. Most eye-tracking studies rely on the eye-mind hypothesis which states that users fixate on an area that relates to the currently processed information. However, special care should be taken when analyzing areas of interest close to each other. Pilots can perceive some information in peripheral visions, for example, speed changing via the movement of the speed tape [74]. The experts may succeed in maintaining a constant speed by looking only at the attitude zone. This would explain why the "AOI SPD" corresponding to speed tape is not often found in the most frequent patterns (n-grams). Finally, we should also specify that eye tracking allow capturing only overt attention, for example when a person moves his eyes in the direction of an object, and not covert attention, when an individual focus his attention on an object, but without moving the eyes toward that object.

## Conclusion

This work highlighted the difference between novices and expert pilots concerning visual scanning strategies and flight performances. Our result confirmed that expertise exerts a top-down modulation on gaze behaviour [10]. We used a wide variety of standard and advanced metrics to uncover the modification of the gaze behavior bring by expertise. Expert pilots have a more efficient perception of the information, better dispersion of their attention, and more elaborate visual patterns. Expertise makes it possible, despite a dual-task costly in visuo-attentional resources, to maintain visual patterns linked with the flying task (i.e. the irrelevant dual-task did not alter the nominal visual behavior). Overall, the eye metrics used in this research are relevant to finely assess pilot's gaze behavior in the cockpit and can contribute to better characterize visual scanning in the cockpit, an important topic for safety [75]. These eye metrics can be used to evaluate pilots during their training program. For example, it might be possible to follow the evolution of their scanning strategies and determine whether they tend to resemble that of expert pilots. In the future, it might be possible to assess cockpit monitoring during real flight [76, 77]. In this sense, recent studies investigated the possibility to use an eye tracking assistant to warn pilots using a database of the visual behaviour of expert pilots [78–80]. Our results suggest that such on-board eye tracking could be customized based on pilot experience. Finally, we believe that the eye metrics employed in this study can be also useful for practitioners and researchers in other fields such as air traffic control and automotive.

## Supporting information

**S1 Fig. Time spent gazing outside the defined AOI for each participants during all scenarios.**
(TIF)

**S1 Data.**
(XLSX)

**S2 Data.**
(XLSX)

**S3 Data.**
(XLSX)

**S4 Data.**
(XLSX)

**S5 Data.**
(XLSX)

**S6 Data.**
(XLSX)

## Acknowledgments

We are especially grateful to all the pilots who participated in the experiment, and to Kevin J. Verdière, Emilie S. Jahanpour, Quentin Chenot, Evelyne Lepron, and Lili for their valuable assistance. The authors thank the PEGASE simulator technical team. The manuscript was improved thanks to three anonymous reviewers.

## Author Contributions

**Conceptualization:** Christophe Lounis, Vsevolod Peysakhovich, Mickaël Causse.

**Data curation:** Christophe Lounis.

**Formal analysis:** Christophe Lounis.

**Funding acquisition:** Christophe Lounis.

**Investigation:** Christophe Lounis, Mickaël Causse.

**Methodology:** Christophe Lounis, Vsevolod Peysakhovich, Mickaël Causse.

**Project administration:** Vsevolod Peysakhovich.

**Resources:** Christophe Lounis.

**Software:** Christophe Lounis.

**Supervision:** Vsevolod Peysakhovich, Mickaël Causse.

**Validation:** Mickaël Causse.

**Visualization:** Christophe Lounis.

**Writing – original draft:** Christophe Lounis.

**Writing – review & editing:** Christophe Lounis, Mickaël Causse.

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
