## [Decision Letter · Decision Letter 0]

11 Sep 2020

PONE-D-20-21444

Visual scanning strategies in the cockpit and pilot expertise: a flight simulator study

PLOS ONE

Dear Dr. LOUNIS,

Thank you for submitting your manuscript to PLOS ONE. After careful consideration, we feel that it has merit but does not fully meet PLOS ONE’s publication criteria as it currently stands. Therefore, we invite you to submit a revised version of the manuscript that addresses the points raised during the review process.

As you will see, the Reviewer's made many interesting and important comments that I won't repeat all here. I believe that the most important aspects to address are: improving the clarity and level of detail; discussing surprising and unexpected findings; and controlling for the fact that the AOIs are of different sizes (this needs a good discussion as to what method you choose, for psychological and practical purposes - using a area-normalised technique might be useful to understand the importance of particular areas psychologically, but may not be as important practically). Potentially, you might want to include multiple analyses in the results. Whatever you choose to do, ensure it is discussed fully. I look forward to receiving your interesting work.

We look forward to receiving your revised manuscript.

Kind regards,

Peter James Hills, PhD

Academic Editor

PLOS ONE

Journal Requirements:

Reviewers' comments:

Reviewer's Responses to Questions

**Comments to the Author**

1. Is the manuscript technically sound, and do the data support the conclusions?

Reviewer #1: Yes

Reviewer #2: Yes

Reviewer #3: Yes

2. Has the statistical analysis been performed appropriately and rigorously? 

Reviewer #1: Yes

Reviewer #2: Yes

Reviewer #3: Yes

3. Have the authors made all data underlying the findings in their manuscript fully available?

Reviewer #1: Yes

Reviewer #2: Yes

Reviewer #3: Yes

4. Is the manuscript presented in an intelligible fashion and written in standard English?

Reviewer #1: Yes

Reviewer #2: No

Reviewer #3: No

5. Review Comments to the Author

Reviewer #1: The article „Visual scanning strategies in the cockpit and pilot expertise: a flight simulator study“ is written at a good level and meets the requirements for a scientific contribution. The topic of the manuscript is relevant and of interest to the audience of this journal. Research methodology and treatment for the study are appropriate and replied properly. Manuscript contain enough sufficient and appropriate references. The level of English is at a high level. The conclusion summarizes the main results and contributions of the manuscript.

Page 7, Eye tracking data „ only AOI- based data were used in this experiment “. What are other areas that are not part of the AOI? (overhead panel, throttle…)

Page 8, Flight simulator data: „In this experiment, the predicted values correspond to the different specific threshold given by the experimenter (i.e., speed 130 Kt; vertical speed below -500 ft/min and above +800 ft/min; heading different from 143°)“ Based on what did you choose these values?

Page 11, Visual patterns identification, article compare tri- gram pattern OTW-VS-OTW, OTW-ECAM-OTW, etc. This marking is opaque. I recommend using a numerical marking according to the distribution of AOI Fig. 2. Overview of the ten different AOIs

I recommend publishing the article in the journal.

Reviewer #2: Manuscript ID: PONE-D-20-21444

Title: "Visual scanning strategies in the cockpit and pilot expertise: a flight simulator study"

This is a nice study dealing with scanning strategies among aviators. The study is well-done and the methods are sound. Future studies should address more specific questions and measurements, for instance whether small saccades produced during fixation (i.e. microsaccades; see McCamy et al., 2014 for a work that discusses the relationship between scene informativeness and ocular targeting), also differ between expert and novice. However, my feeling is that the current study already makes a valuable contribution to the field in present form.

Notwithstanding the above, I have some comments and suggestions.

Introduction

#1. The literature reviewed in the introduction and discussion seems incomplete. When revising your paper, please take care to ensure your reference list is up to date, and that any recent paper that are of relevance to your work are cited.

#2. I suggest adding a table comparing the eye movement metrics you described (e.g. main differences, shortcomings, strength, etc…). It will tremendously help the reader to have a big picture of your arguments.

Methods

#3. I suggest adding more details (or better defining) of your novice group. It looks quite strange that participants with no flight experience were able to flight an aircraft. Did they receive any basic flight training? If they have received some basic flight courses during their education in aeronautics, it should be stated (number of hours). It will help to strengthen your results and discussion. If this group lacks of minimum flight notions on how to interact with the aircraft, it should be reported as a main shortcoming of your study. In this case, any comparisons with the expert group will be pointless, and any mention (including the title) comparing novices and experts should be toned down.

#4. Please, add more details about the choice of your sample size. You might want to state if the number of pilots was considered appropriate based on a previous cohort.

#5. What is the field of view covered by the simulator?

#6. The AOIs are very different in size. Is this controlled for in any way? Furthermore, AOIs 6 and 7 are adjacent to each other and are relatively small, at least in terms of the subtended visual field. For example, if a pilot is looking at 6, do he really need to foveate 7 to get the necessary information. Moreover, can your eye tracking system resolve these different AOIs reliably? You might want to discuss this issue in your limitation section.

#7. Did you re-calibrate the 5 cameras, and eventually update the eye tracker 3D setting, for each pilot? Please clarify this issue.

#8. Authors stated that have used machine learning models. Which models? In plural?

Results

#9.It is surprising that participants with no flight experience (novices) were able to behave similar to the expert pilots (i.e. Heading ). This supports my comment on the description of your sample. Furthermore, it feels strange that novices and pilots behave the same during the easy dual task scenario, but very differently in the other two. Is there any plausible explanation for this?

Discussion

#10. I suggest providing a theoretical framework for describing your results. While this is not absolutely necessary to publish the results, it will strengthen the argument about differences in visual strategy.

Suggested references:

Shiferaw, B., Downey, L., & Crewther, D. (2019). A review of gaze entropy as a measure of visual scanning efficiency. Neuroscience & Biobehavioral Reviews, 96, 353-366.

Diaz-Piedra, C., Rieiro, H., Cherino, A., Fuentes, L. J., Catena, A., & Di Stasi, L. L. (2019). The effects of flight complexity on gaze entropy: An experimental study with fighter pilots. Applied ergonomics, 77, 92-99.

McCamy, M. B., Otero-Millan, J., Di Stasi, L. L., Macknik, S. L., & Martinez-Conde, S. (2014). Highly informative natural scene regions increase microsaccade production during visual scanning. Journal of neuroscience, 34(8), 2956-2966.

Reviewer #3: The manuscript presents an experiment that explored the potential of several eye metrics to differentiate between pilots with different flight expertise. It presents really interesting, innovative metrics (k coefficient, transition entropy, n-gram coding), although authors do not really justify the selection of such metrics and the classification of metrics seems forced (most metrics look to me more as gaze patterning metrics than gaze dispersion metrics). Anyway, the methods used are generally sound and valid and I feel that the manuscript will be of interest and very useful for researchers in the field.

More detailed comments are provided below.

1. The manuscript needs to be good, thoughtful proofread. Also, the messages are sometimes not clear and therefore the “story” that authors are telling is hard to follow. I am not only proposing authors to remove typos and spelling errors, or to carefully edit the whole manuscript for correctness and clarity, but to improve the organization of the contents. Some errors, only from the abstract section, are as follows. Some of them should have been detected prior submission as they clearly affect the flow of the text.

- The first sentence is written in a way that simplifies the construct of situational awareness: here, situational awareness would merely represent the monitoring of flight instruments. Moreover, monitoring the flight instruments is not the only demanding activity in the cockpit, and just monitoring them does not guarantee a timely intervention in an emergency situation.

- I know that there are texts that use the term “situation awareness”, whereas others use “situational awareness”. In this case, authors should only choose one term and stick to it.

- It is not clear what authors mean by “qualify” visual strategies or visual information in the abstract. It needs to be clarified if authors are referring to “study”, or to “examine”, or any other synonym, or maybe they are referring to “quantify”.

- Does “visual information taking” mean “visual information acquisition”?

- Does “efficient perceptual efficiency” mean “higher perceptual efficiency”?

- The sentence “The two groups performed ...” in the abstract should be rephrased as its structure does not comply with English grammar rules.

- “complex” and “elaborate” are essentially synonyms. What does “elaborate” add to the sentence?

- Authors mentioned “better dispersion of their (pilots) attention” in the abstract. Attention is a very complex construct. What authors are referring to with the idea of a dispersed attention? I understand that they are probable referring to orienting, a kind of attention triggered by external cues (visual or in other modalities) and that usually implies the movements of the eyes toward a target location. It is important that authors differentiate between overt and covert orienting attention, as the latter may be especially relevant for expert pilots.

- The sentence “These visual scanning differences…” in the abstract should be rephrased. The scanning differences are being used to classify pilots, this reads like the other way around.

- The sentence “Our results can benefit…” is saying the same thing twice. Also, “benefit for aviation” does not make sense. I understand it would be “benefit the aviation”.

2. Authors presented a classification of gaze patterns depending on “spatial dispersion” and “its structure”. I am not sure why the “gaze dispersion metrics” are called that way. For example, the transition matrix includes a significant amount of temporal structure, and the k coefficient, while it somehow includes dispersion (saccade size), it is influenced at least 50% by dwell time, which again is a temporal parameter. Also, it is unclear why the transition entropy is not included here, since it is a very similar measure to the transition matrix density. In general, I am not sure there is a clear distinction between the two set of parameters, but, if any, the comparison is more between global and fine level structure.

Regarding the transition matrix density, authors stated that “A sparse matrix (small index value) indicates a more efficient and directed search”. It is not necessarily more efficient, since it can be a marker of missing vital information. For example, a novice driver can direct his gaze continuously to the road while ignoring/forgetting the rearview mirrors.

Regarding the k coefficient, authors stated that “Values of Ki close to zero indicate relative similarity between dwell durations and transition amplitudes.” It would be interesting to know if this comes from long dwelling periods followed by large saccades or short dwells and small saccades, since they are probably generated by very different cognitive/physiological states.

Finally, there is an analysis using n-grams that is used on the data but not described here, which is extremely confusing.

3. Methods:

- Authors defined the flight scenarios used, but never the abbreviations used across the rest of the manuscript.

- As far as I can tell, there is not a catch-all AOI being defined (i.e., subject is looking somewhere the design did not account for). Is there a significant amount of time spent looking outside these AOIs?

- The eye tracking data section is probably the most important on the manuscript and it is very hard to understand. Authors should rewrite it carefully to improve readability. Moreover:

1) Authors defined two types of entropy in the introduction section, but did not clarify the one they finally used.

2) Pattern identification has not been explained before.

3) Machine learning models should be described.

4) A potential thing that could be added is an analysis on metric redundancy. Metrics such as the transition matrices and the transition entropy seem to be measuring very similar things, and kind of the same with the LZC and the ngrams. Is it really useful to have all of them?

4. Results:

- Figure 9 differences in transition matrices between expert and novice pilots. Which one has the more homogeneous distribution? Authors need to state it in the results section. Also, it seems like most of the novice complexity comes from exploration of AOIs 1-5. Is this relevant?

- The true positive-false positive seems redundant, since it has the exact same information as the confusion matrix.

- The focal-ambient K coefficient showed that attention was dominantly focal (positive value) in both groups, but not in all scenarios. Is that correct?

- In the hard dual-task scenario, authors found that dual-task changed the ambient-focal strategy of the novices, while experienced pilots kept their strategy consistent across experimental scenarios. Any idea as to why?

6. PLOS authors have the option to publish the peer review history of their article (what does this mean?). If published, this will include your full peer review and any attached files.

Reviewer #1: No

Reviewer #2: No

Reviewer #3: No

---

## [Author Response · Author response to Decision Letter 0]

2 Dec 2020

Please find all the responses in the document "response to reviewer.pdf".

Reviewer #1

The article „Visual scanning strategies in the cockpit and pilot expertise: a flight simulator study“ is written at a good level and meets the requirements for a scientific contribution. The topic of the manuscript is relevant and of interest to the audience of this journal. Research methodology and treatment for the study are appropriate and replied properly. Manuscript contain enough sufficient and appropriate references. The level of English is at a high level. The conclusion summarizes the main results and contributions of the manuscript. Page 7, Eye tracking data „ only AOI- based data were used in this experiment “. What are other areas that are not part of the AOI? (overhead panel, throttle…)

• Thank you very much for these positive and kind comments. We indeed did not integrate these AOIs as our experiment specifically addressed the monitoring issues during the landing phase. Consequently, we restricted our investigation to the flight instruments that displayed relevant information during this maneuver. The flight scenarios were designed in such a way that no action was required on the overhead panel. However, the authors recognize that we could have analyzed the time spent gazing at the throttle, as the position of the thrust levers can be monitored by crews. However, we choose to restrict our analysis to instruments that display information directly related to the flight parameter (altitude, speed, etc.). We clarified this point in the paper.

Page 8, Flight simulator data: „In this experiment, the predicted values correspond to the different specific threshold given by the experimenter (i.e., speed 130 Kt; vertical speed below -500 ft/min and above +800 ft/min; heading different from 143°)“ Based on what did you choose these values?

• These values were chosen because they roughly correspond to reality. In commercial aircraft, a standard landing speed is around 130 kt. The chosen negative vertical speed (–500 ft/min) roughly corresponds to the vertical speed during the approach at 130 kt, with an angle of approach of three degrees. We defined a tolerance range in case the participant was not well stabilized on the approach slope. Consequently, we gave these instructions to our participants to perform their landing. We added these details in the paper: 

o “We choose these values because they roughly correspond to a standard landing speed with a commercial aircraft. The negative vertical speed of 800 ft/min approximatively corresponds to the vertical speed at 130 kt with an angle of approach of three degrees. We defined a tolerance range in case the participant was not well stabilized on the approach slope and had to gain altitude (+500 ft/min maximum).”

Page 11, Visual patterns identification, article compare tri- gram pattern OTW-VS-OTW, OTW-ECAM-OTW, etc. This marking is opaque. I recommend using a numerical marking according to the distribution of AOI Fig. 2. Overview of the ten different AOIs

• We thank reviewer #1 for this suggestion. The author would like to keep the labels (OTW, VS, SPD...) since they are commonly used in experiment involving flight instruments. If reviewer #1 feels that numbers could clarify, we might add the numerical mark in parentheses in addition to the label.

Reviewer #2

#1. The literature reviewed in the introduction and discussion seems incomplete. When revising your paper, please take care to ensure your reference list is up to date, and that any recent paper that are of relevance to your work are cited.

• We expanded the introduction and discussion section by adding recent supplementary recent references relevant to our work. In particular, the following references were introduced: Diaz-Piedra et al. (2019) ; Shiferaw et al. (2019) ; Haslbeck et Zhang (2017) ; Allsop (2014) ; Brams et al. (2018) ; Alamán et al. (2020),Dubois et al. (2017), Schwerd et Schulte (2020), Peissl et al(2019).

#2. I suggest adding a table comparing the eye movement metrics you described (e.g. main differences, shortcomings, strength, etc…). It will tremendously help the reader to have a big picture of your arguments.

• We thank reviewer #2 for this excellent suggestion that helped clarify the presentation of the various eye metrics. We add a table comparing the different metrics.

#3. I suggest adding more details (or better defining) of your novice group. It looks quite strange that participants with no flight experience were able to flight an aircraft. Did they receive any basic flight training? If they have received some basic flight courses during their education in aeronautics, it should be stated (number of hours). It will help to strengthen your results and discussion. If this group lacks minimum flight notions on how to interact with the aircraft, it should be reported as a main shortcoming of your study. In this case, any comparisons with the expert group will be pointless, and any mention (including the title) comparing novices and experts should be toned down.

• All novices’ participants had advanced theoretical knowledge about aeronautical engineering, and they all knew perfectly the various information given by the instruments in the cockpit (altimeter, altitude etc.), and all had flight notions on how to manually interact with the aircraft. However, they had no experience of real flying. In our experiment, the participant simply had to control the trajectory and the speed of the aircraft, thus the task was feasible by beginners (the scenarios did not require complex navigation activities or interaction with automation). However, the authors acknowledge that assessing different levels of expertise, including pilots with low, moderate, and high experience, would be a desirable extension of this work. We clarified this point in the participant section:

o “A first group called “novices” consisted of participants with no real flight experience (n = 16, mean age 25.65 ± 5.47 years). They were recruited from a French aerospace engineering school (ISAE-SUPAERO, Toulouse, France). All these novices participants had advanced theoretical knowledge about aeronautical engineering, were familiar with the various information given by the instruments in the cockpit (altimeter, altitude etc.), and had flight notions on how to manually interact with the aircraft. Our experimental flight scenarios were relatively simple: the participant had to control the trajectory and the speed of the aircraft. The scenarios did not require complex navigation activities or interacting with automation. Thus the scenarios were feasible for these novices after a relatively short training session.”

• They indeed all went thought a training session, we slightly expanded this point:

o “Participants performed a training session, consisting of performing two times a landing scenario. All participants (including novice ones) were able to control the aircraft correctly after these two landings. Then, the participants performed three times the same landing scenario than during the training, but with varying levels of complexity.”

• Finally, we expanded the discussion on this possible shortcoming in the limitation section as follow:

o “There are some limitations to this study. We compared professional pilots with non-pilots only. The comparison of these two very different profiles can artificially increase the observed differences in terms of ocular behavior. A further researcher should consider participants with different levels of expertise from novice to expert (e.g., every 1000 hours) to finely examine the implementation of the visual strategies with expertise.”

#4. Please, add more details about the choice of your sample size. You might want to state if the number of pilots was considered appropriate based on a previous cohort.

• With a total of 32 participants, divided in two groups (16 novices and 16 experts), our study is in the range of other works. Some studies involved a more reduced sampled size (e.g., 10 novices and 10 pilots for Ottati et al. (1999); 10 novices and 6 experts’ pilots for kasarskis (2001)); 14 novices and 14 pilots for Schriver et al (2008). Other studies involved a relatively similar sample size: 36 pilots for Wen-Chin Li et al. (2012) with various flight hours experiences. In order to determine a sufficient sample size, we initially conducted a statistical power analysis to determine the sample size required to detect an effect with power = 0.9 as a function of standardized effect size (alpha = 0.05). We compared average dwell times on all cockpits AOI for 5 novices and 5 experts that conducted a pretest scenario. This power analysis showed that approximately 10 participants per group were sufficient to reach a power of 0.90.

  

#5. What is the field of view covered by the simulator?

• The field of view covered by the simulator is 180°. This information has been added in the paper

#6. The AOIs are very different in size. Is this controlled for in any way? Furthermore, AOIs 6 and 7 are adjacent to each other and are relatively small, at least in terms of the subtended visual field. For example, if a pilot is looking at 6, do he really need to foveate 7 to get the necessary information. Moreover, can your eye tracking system resolve these different AOIs reliably? You might want to discuss this issue in your limitation section.

• Reviewer #2 is right. Indeed, AOIs are of different sizes since they correspond to the size of the various instruments. We did not control these size variations since we believe that this has no particular impact on our analysis as AOIs are the same for novice and experts. However, we fully acknowledge that relatively small and close AOIs can bias results due to the possibility to process information in peripheral vision and also due to the accuracy limitations of eye tracker systems. However, regarding AOI 6 and 7, we believe that it was not problematic since the distance between the center of the ND and the values in the AOI 7 are relatively far. Also, when defining the AOI 7, being aware of the accuracy limitations of eye tracking systems, we took a slight margin around the AOI 6 in order to ensure that dwell on AOI 7 would be captured. Indeed, there is no other reason for the pilot to glance close to that location, except reading the values. We expended discussion about these issues in the limitation section as follow: 

o “Most eye-tracking studies rely on the eye-mind hypothesis which states that users fixate an area that relates to the currently processed information. However, special care should be taken when analyzing areas of interest close to each other. Pilots can perceive some information in peripheral visions, for example, speed changing via the movement of the speed tape (Citer : Alamán, J. R., Causse, M., & Peysakhovich, V. (2020). Attentional span of aircraft pilots: did you look at the speed? In 1st International Workshop on Eye-Tracking in Aviation). The experts may succeed in maintaining a constant speed by looking only at the attitude zone. This would explain why the ”AOI SPD” corresponding to speed tape is not often found in the most frequent patterns (n-grams). Finally, the eye tracker devices are more and more mature and accurate (about 1°at a distance of one meter). However, care should be taken when analyzing contiguous AOIs, the limitation in accuracy of eye tracking systems could lead to errors in this situation.”

#7. Did you re-calibrate the 5 cameras, and eventually update the eye tracker 3D setting, for each pilot? Please clarify this issue.

• Smart eye system allows to design a 3D environment and to establish calibration points (at the vicinity of AOIs). When the world model is designed, we just need to operate an automatic calibration for each participant.

#8. Authors stated that have used machine learning models. Which models? In plural?

• We used different classification algorithms (support vector Machine, LDA, Decision trees, ...) to determine which algorithm performed the best accuracy. The algorithm performing better was used in this paper to classify expertise based on transition matrix. 

Results

#9.It is surprising that participants with no flight experience (novices) were able to behave similar to the expert pilots (i.e. Heading ). This supports my comment on the description of your sample. Furthermore, it feels strange that novices and pilots behave the same during the easy dual task scenario, but very differently in the other two. Is there any plausible explanation for this?

• This is an excellent remark. The heading metric was not really sensitive to expertise since the aircraft was nearly in front of the airfield at the beginning of the scenario. Consequently, it was relatively trivial to keep the aircraft to the correct heading. Speed and vertical speed values were much more complex to maintain, explaining why pilot performed systematically better than novices on these two variables. We clarified this point in the paper:

o “Our results showed that experts’ pilots had better flying performance than novices. In particular, they had lower speed and vertical speed deviations (heading variable was not sensitive most likely because the aircraft was nearly in front of the airfield at the beginning of the scenario). We assume that these superior flying performances are at least partially due to better visual scanning strategies gained with expertise.”

Discussion

#10. I suggest providing a theoretical framework for describing your results. While this is not absolutely necessary to publish the results, it will strengthen the argument about differences in visual strategy.

• Thank you very much for this excellent suggestion. We added a section in discussion referring to three possible theoretical models that can account for our results, in particular regarding the differences between novices and experts.

o Three theories can explain} the expert superiority in visual domains. First, the theory of long-term working memory \\cite{bib70} assumes that expertise extends the capacities for information processing. This theory assumes that the limited-capacity assumption should be reconsidered when related to an expert’s specific domain. Related to this hypothesis, experts encode and retrieve information more rapidly than novices. This expert's rapid information processing is reflected in shorter dwell durations. The second theory is related to the information-reduction hypothesis \\cite{bib71}. This assumes that expertise optimizes the amount of processed information by neglecting task-irrelevant information. Our results demonstrated that expert’ group keeps maintaining the visual scanning strategies related to the piloting activity during the hard-dual task scenario while novice under-performed during this scenario. This result highlights the expert's ability to focus toward relevant information to perform the task neglecting redundant information. Eventually, the third theory is the holistic model of image perception \\cite{bib72}. It focuses on the extension of the visual span. Charness et al. \\cite{bib73} shown that experts extract information from widely distanced and parafoveal regions, producing patterns of saccadic selectivity by piece saliency \\cite{bib74}. Our results suggested that expert over-performed the novice group in maintaining their speed. N-gram analysis revealed the visual scanning strategies related to speed were not found for the pilots whereas novices presents this AOI in their sequences. These results suggested the ability for experts to process information through parafoveal processing.

Reviewer #3

The manuscript presents an experiment that explored the potential of several eye metrics to differentiate between pilots with different flight expertise. It presents really interesting, innovative metrics (k coefficient, transition entropy, n-gram coding), although authors do not really justify the selection of such metrics and the classification of metrics seems forced (most metrics look to me more as gaze patterning metrics than gaze dispersion metrics). Anyway, the methods used are generally sound and valid and I feel that the manuscript will be of interest and very useful for researchers in the field.

More detailed comments are provided below.

1. The manuscript needs to be good, thoughtful proofread. Also, the messages are sometimes not clear and therefore the “story” that authors are telling is hard to follow. I am not only proposing authors to remove typos and spelling errors, or to carefully edit the whole manuscript for correctness and clarity, but to improve the organization of the contents. Some errors, only from the abstract section, are as follows. Some of them should have been detected prior submission as they clearly affect the flow of the text.

• We proofread the manuscript and we separated the beginning of the manuscript into two sections "Introduction" and "State-of-the-art visual scanning metrics" to facilitate the reading. We also enhanced the text flow to improve the reading experience.

The first sentence is written in a way that simplifies the construct of situational awareness: here, situational awareness would merely represent the monitoring of flight instruments. Moreover, monitoring the flight instruments is not the only demanding activity in the cockpit, and just monitoring them does not guarantee a timely intervention in an emergency situation.

• We do agree with the reviewer #3, we mitigated the first sentence as follow:

o “During a flight, pilots must rigorously monitor their flight instruments since it is one of the critical activities that contribute to update their situation awareness. This task is cognitively demanding, but it is necessary for timely intervention in the event of a parameter deviation.”

I know that there are texts that use the term “situation awareness”, whereas others use “situational awareness”. In this case, authors should only choose one term and stick to it.

• We homogenized the term and now use only the term “situation awareness”

It is not clear what authors mean by “qualify” visual strategies or visual information in the abstract. It needs to be clarified if authors are referring to “study”, or to “examine”, or any other synonym, or maybe they are referring to “quantify”.

• We replaced the term by “study “

 Does “visual information taking” mean “visual information acquisition”?

• We replaced the term by “visual information acquisition, which is more widely used”.

 Does “efficient perceptual efficiency” mean “higher perceptual efficiency”?

• We replaced by “higher perceptual efficiency”

 The sentence “The two groups performed ...” in the abstract should be rephrased as its structure does not comply with English grammar rules.

• Thanks, the sentence has been rephrased as follow:

o “The two groups landed three times with different levels of difficulty (manipulated via a double task paradigm)”.

“complex” and “elaborate” are essentially synonyms. What does “elaborate” add to the sentence?

• We believe that “elaborate” brings here information suggesting the beneficial result of experience and is thus more positively connoted than “complex”. We can remove this word if reviewer #3 feels that it does not bring any additional information than “complex”.

Authors mentioned “better dispersion of their (pilots) attention” in the abstract. Attention is a very complex construct. What authors are referring to with the idea of a dispersed attention? I understand that they are probable referring to orienting, a kind of attention triggered by external cues (visual or in other modalities) and that usually implies the movements of the eyes toward a target location. It is important that authors differentiate between overt and covert orienting attention, as the latter may be especially relevant for expert pilots.

• We intended to refer to the idea that visual attention was more spatially distributed to the visual field (i.e. in more equal proportion vs. concentrated at few area of interest). In other words, their attention was not focused on a single channel of information (phenomenon called attentional tunneling when extreme, cf. Wickens 2005, Attentional Tunneling and Task Management). In the diffuse mode, visual attention is allocated to all regions of the visual field in equal proportion; in the focused mode, attention is concentrated at one area of interest, specified by a central or peripheral cue. (citer) Heitz, R. P., & Engle, R. W. (2007). Focusing the spotlight: Individual differences in visual attention control. Journal of Experimental Psychology: General, 136(2), 217. 

• We replaced the term by "better distribution of attention” to the cockpit instruments and we slightly developed the notions of diffuse vs. focused attention as well as attentional tunneling concept in the attentional modes section: 

o “According to Heitz et al. (2007), in the diffuse mode, visual attention is allocated to all regions of the visual field in equal proportion; in the focused mode, attention is concentrated at one area of interest, specified by a central or peripheral cue. An extremely focused mode could be compared to the concept of attentional tunneling (Wickens et al. 2009).”

We added in limitation overt vs covert orienting attention concerning eye-tracking device.

o Finally, we should also specify that eye tracking allow capturing only overt attention, for example when a person moves his eyes in the direction of an object, and not covert attention, when an individual focus his attention on an object, but without moving the eyes toward that object.

The sentence “These visual scanning differences…” in the abstract should be rephrased. The scanning differences are being used to classify pilots; this reads like the other way around.

• The reviewer is right we replaced the sentence by:

o We classified pilot's profiles (novices -- experts) by machine learning based on Cosine KNN (K-Nearest Neighbors) using transition matrices.

The sentence “Our results can benefit…” is saying the same thing twice. Also, “benefit for aviation” does not make sense. I understand it would be “benefit the aviation”.

• Reviewer 3 is right ; we make the necessary revisions in the text concerning the points mentioned below to improve readability. 

o “Our results benefit the aviation domain to speed up learning, assess the monitoring performance of the crew, and ultimately reduce incidents or accidents due to human errors.

2. Authors presented a classification of gaze patterns depending on “spatial dispersion” and “its structure”. I am not sure why the “gaze dispersion metrics” are called that way. For example, the transition matrix includes a significant amount of temporal structure, and the k coefficient, while it somehow includes dispersion (saccade size), it is influenced at least 50% by dwell time, which again is a temporal parameter. Also, it is unclear why the transition entropy is not included here, since it is a very similar measure to the transition matrix density. In general, I am not sure there is a clear distinction between the two set of parameters, but, if any, the comparison is more between global and fine level structure.

• Reviewer #3 is right, we struggled to propose a clear classification, but we agree that it does not described in a sufficiently accurate fashion the specificity of each metric. We proposed a new classification in the text and a table is now introduced to describe the different metrics. We hope that this classification is more accurate.

o One approach to analyze visual scanning strategies is to analyze transition matrix, a second one is the characterization of fluctuation between ambient/focal visual behavior}, another one is to derive global patterns metrics such as entropy. More generally, in this paper, we classified visual scanning strategies metrics in three AOI based approaches: one is based on Markov chains (transition matrix), another is based on the attentional modes, and the last one is based on sequences analyses}. Figure 14 presents a comparison of the visual scanning metrics described below (e.g. formula, definition, strength shortcomings, strength, etc.).” 

Regarding the transition matrix density, authors stated that “A sparse matrix (small index value) indicates a more efficient and directed search”. It is not necessarily more efficient, since it can be a marker of missing vital information. For example, a novice driver can direct his gaze continuously to the road while ignoring/forgetting the rearview mirrors.

• Reviewer #3 is absolutely right; this sentence has been improved. We mentioned now the fact that a sparse matrix can reflect a more efficient and directed search, for example when using a computer software (Goldberg, 2002), or, in other contexts, can indicate a failure to properly monitor the environment, for example when a novice driver directs his gaze continuously to the road while ignoring/forgetting the rearview mirrors or when a pilot is excessively engaging his visual attention on a channel of information (e.g., Wickens 2009). The following sentence was introduced: 

o “A sparse matrix can reflect a more efficient and directed search, for example when using a computer software (Goldberg,2002), or, in other contexts, can indicate a failure to properly monitor the environment, for example when a novice driver directs his gaze continuously to the road while ignoring/forgetting the rearview mirrors or when a pilot is excessively engaging his visual attention on a single instrument (e.g., Wickens 2009).”

Regarding the k coefficient, authors stated that “Values of Ki close to zero indicate relative similarity between dwell durations and transition amplitudes.” It would be interesting to know if this comes from long dwelling periods followed by large saccades or short dwells and small saccades, since they are probably generated by very different cognitive/physiological states.

• The reviewer #3 is right future study should investigate this question we add.

o It is worth noting that the values of K coefficient should be interpreted together with dwell duration results because different groups can have different average values of dwell duration and transition amplitudes.}

Finally, there is an analysis using n-grams that is used on the data but not described here, which is extremely confusing.

• To the author's knowledge, this is the first time that the n-gram comparison is used in a study involving eye tracking data, thus we could not cite any reference and the n-gram analysis was described in the section method/data processing only. However, we do agree with reviewer #3 that a description was needed, this metric is now depicted in the table.

3. Methods:

Authors defined the flight scenarios used, but never the abbreviations used across the rest of the manuscript.

• The abbreviations concerning the flight scenarios were defined in figure label. But reviewer #3 is right and we have now added the abbreviations also in method section to ease the reading. 

o Participants manually (i.e., without the autopilot) performed three times the same landing scenario according to three different conditions. The “control scenario” (CS) was a nominal landing without a supplementary task. The “easy dual task scenario” (EDTS) and the “difficult dual task scenario” (HDTS) were similar to the “control scenario” except that participants were asked to perform a supplementary monitoring task.

As far as I can tell, there is not a catch-all AOI being defined (i.e., subject is looking somewhere the design did not account for). Is there a significant amount of time spent looking outside these AOIs?

This is a very good question. We ensure that there is not a significant amount of time spent looking outside an AOI. The graph below shows the percentage of time spent outside an AOI for each participant numbered from 1 to 16. We added this graph in supplementary materials.

The eye tracking data section is probably the most important on the manuscript and it is very hard to understand. Authors should rewrite it carefully to improve readability.

• This section has been rewritten to improve clarity and readability.

o Figure 3 shows the entire eye tracking pipeline analysis. Each AOI was coded using numbers from 1 to 10 corresponding to the flight instruments (see \\ref{Fig3}). Only AOI-based data were extracted in this experiment and concatenated to obtain two chronological vectors containing the indices of the visited AOIs (from 1 to 10) and the time spent on them. Dwells inferior to 200 ms \\cite{bib53} were discarded. Furthermore, consecutive fixations in the same area were merged (e.g., for 1, 1, 4, 4, 5, 5, 5, 6 we only consider 1, 4, 5, 6). The transition vector (the vector containing the transitions between each AOI numbers) was used to compute LZC, GTE. Concerning the transition matrices, given their high dimensionality, it is difficult to use classical inferential statistics. Therefore, we applied machine learning algorithms on the concatenated transition matrices to compare the two groups of participants (novice vs pilot). \\hl{Various machine learning model types were used (SVM, LDA, K-Nearest Neighbor, for a review see} \\cite{bib67}). \\hl{The algorithm performing the best accuracy (Cosine KNN) was selected in this paper}. The transition probabilities from one AOI to another were taken as a feature, thus raising the number of features to a total of 100 features (i.e., 10 AOIs × 10 AOIs). A principal component analysis (PCA) was used to reduce the features’ numbers. This restricts the model to 35 features corresponding to the main transition probabilities of the matrices. Five-fold cross-validation was used, which is a good trade-off between bias and variance estimation \\cite{bib54}. According to Combrisson and Jerbi \\cite{bib55} theoretical chance level for classification for $p$ $<$ 0.05 with two classes is around 58\\%. Concerning the K coefficient, the transition entropy, and the Lempel Ziv complexity methods, they were respectively computed following the methods of \\cite{bib44}, \\cite{bib50}, \\cite{bib52}. Finally, based on the transition vector, the n-grams frequency-based method \\cite{bib56} was used to identify the number of common 3, 4, 5, and 6-gram sequences in each group. After counting the occurrence of given n-grams for each participant, the number of common sequences of each n-gram was calculated for each group (Novice/Pilots) .

o 

1) Authors defined two types of entropy in the introduction section but did not clarify the one they finally used.

• We agree that this is a useless confusion. We have clarified this point in the text by removing from the text the entropy type that is not used in this paper.

2) Pattern identification has not been explained before.

• We have clarified this point by adding supplementary elements describe in introduction and method:

o In introduction: N-gram is an essential component of many methods in bioinformatics, including genome and transcriptome assembly, metagenomic sequencing, and error correction of sequence reads} \\cite{bib66}. Basically, an N-gram model might predict the occurrence of an AOI, based on the occurrence of its N–1 previous AOI. So here, we are answering the question: how far back in the history of a sequence of AOI should we go to predict the next AOI? For instance, a bigram model (N=2) predicts the occurrence of an AOI given only its previous AOI (as N–1=1 in this case). Similarly, a trigram model (N=3) predicts the occurrence of an AOI based on its previous two AOIs. The common N-gram sequence analysis uses the n-grams frequency-based method \\cite{bib56} to identify the number of common 3, 4, 5, and 6-gram sequences in each group. By using this method, it is possible to count the occurrence of N-gram AOI and their occurrence for each pilot, and thus, it allows to compare for each N-gram the intra-group patterns consistency.

o In method: The n-grams frequency-based method \\cite{bib56} was used to identify the number of common 3, 4, 5, and 6-gram sequences in each group. After counting the occurrence of given n-grams for each participant, the number of common sequences of each n-gram was calculated for each group (Novice/Pilots).

3) Machine learning models should be described.

• We have added a description concerning Machine learning models:

o “We used different classification algorithms (support vector Machine, LDA, Decision trees, ...) to determine which algorithm performed the best accuracy. The algorithm cosine KNN performing better was used in this paper to classify expertise based on transition matrix.” 

4) A potential thing that could be added is an analysis on metric redundancy. Metrics such as the transition matrices and the transition entropy seem to be measuring very similar things, and kind of the same with the LZC and the n-grams. Is it really useful to have all of them?

• The author thanks reviewer #3 for this very relevant comment. As proposed by another reviewer we add a table comparing the different metrics (shortcoming, strength, formula, approach.) We believe that each metric has its strengths and weaknesses to bring an understanding of visual strategies. We add supplementary explanation in discussion : 

o All the metrics used in this study allowed characterizing visual scanning. We examined the impact of expertise and flying difficulty on the visual scanning strategies. As our results showed, a large number of standard and advanced metrics were sensitive to these two factors. Each metric has its strengths and weaknesses to bring an understanding of visual strategies. For instance, while a transition matrix measure and an entropy value are closely related, the information presented for one and the other is different. A transition matrix makes it possible to measure the preferred paths when consulting AOIs. It highlights the strength of the links between AOIs while the entropy will reflect the disorder of these transition sequences. The application of these metrics can be different. For example, if the aim is to redesign a cockpit panel, transition matrices can be very useful because they give the strength of the relationship between AOIs. This metric can allow to bring close together instruments that are often gazed consecutively, which would help to spare the pilot's visual attention effort. Concerning LZC and N-gram method, N-gram compares the patterns used within the group, while LZC assesses the compressibility of the patterns (how varied the patterns are).

4. Results:

Figure 9 differences in transition matrices between expert and novice pilots. Which one has the more homogeneous distribution? Authors need to state it in the results section. Also, it seems like most of the novice complexity comes from exploration of AOIs 1-5. Is this relevant?

• Reviewer #3 is right; we added other elements in the text to discuss these results. 

o The differences in transition matrices between novices/pilots are mainly observed in a more sparsed distribution of transition probabilities from one instrument to another for the pilot's group. Most of the AOI explored by Novice group involved AOI concentrated in the PFD (from 1 to 5, see Fig3) while pilot’s group explore other combinations of AOI.

The true positive-false positive seems redundant, since it has the exact same information as the confusion matrix.

• This is the current diagram for presenting the results of machine learning. We propose to keep it, but if reviewer #3 is really opposed to it, we can delete it.

The focal-ambient K coefficient showed that attention was dominantly focal (positive value) in both groups, but not in all scenarios. Is that correct? In the hard dual-task scenario, authors found that dual-task changed the ambient-focal strategy of the novices, while experienced pilots kept their strategy consistent across experimental scenarios. Any idea as to why?

• Yes, this is indeed the case. These results and interpretation are discussed in the discussion section. We added more details in discussion section to clarify this point in the text. 

o Our explanation for this result would be that during the easy double task, novices can set up visual strategies to be able to fly the aircraft and perform the double task. As shown in both groups (novices and pilots), the addition of the double task required the participants to look at an extra zone (ND zone) which would be the explanation of the ambient mode for both groups. However, during the difficult double task, due to the time pressure the novices cannot check the value displayed in the ND zone as frequently as they should, which implies more omission error in the difficult double task compared to the easy double task. The consequence of this visual behaviour is that a more focused mode is generated while the pilots manage to maintain their visual strategies under high time pressure.

---

## [Decision Letter · Decision Letter 1]

1 Feb 2021

Visual scanning strategies in the cockpit are modulated by pilots' expertise: a flight simulator study

PONE-D-20-21444R1

Dear Dr. LOUNIS,

We’re pleased to inform you that your manuscript has been judged scientifically suitable for publication and will be formally accepted for publication once it meets all outstanding technical requirements.

Kind regards,

Peter James Hills, PhD

Academic Editor

PLOS ONE

Additional Editor Comments (optional):

Reviewers' comments:

Reviewer's Responses to Questions

**Comments to the Author**

1. If the authors have adequately addressed your comments raised in a previous round of review and you feel that this manuscript is now acceptable for publication, you may indicate that here to bypass the “Comments to the Author” section, enter your conflict of interest statement in the “Confidential to Editor” section, and submit your "Accept" recommendation.

Reviewer #1: All comments have been addressed

Reviewer #2: All comments have been addressed

Reviewer #3: All comments have been addressed

2. Is the manuscript technically sound, and do the data support the conclusions?

Reviewer #1: Yes

Reviewer #2: Yes

Reviewer #3: Yes

3. Has the statistical analysis been performed appropriately and rigorously? 

Reviewer #1: Yes

Reviewer #2: Yes

Reviewer #3: Yes

4. Have the authors made all data underlying the findings in their manuscript fully available?

Reviewer #1: Yes

Reviewer #2: Yes

Reviewer #3: Yes

5. Is the manuscript presented in an intelligible fashion and written in standard English?

Reviewer #1: Yes

Reviewer #2: Yes

Reviewer #3: No

6. Review Comments to the Author

Reviewer #1: (No Response)

Reviewer #2: The authors have addressed all my comments and it was a pleasure to read the revised version of the manuscript.

I am pleased to endorse this manuscript.

Reviewer #3: The authors have done a very thorough job in revising the manuscript and addressing my concerns. I believe that this manuscript is now suitable for publication, although I would urge authors to look for editing help from someone with full professional proficiency in English and to ensure the reference list is up to date.

7. PLOS authors have the option to publish the peer review history of their article (what does this mean?). If published, this will include your full peer review and any attached files.

Reviewer #1: No

Reviewer #2: No

Reviewer #3: No

---

## [Editor Report · Acceptance letter]

3 Feb 2021

PONE-D-20-21444R1 

Visual scanning strategies in the cockpit are modulated by pilots’ expertise: a flight simulator study  

Dear Dr. Lounis:

I'm pleased to inform you that your manuscript has been deemed suitable for publication in PLOS ONE. Congratulations! Your manuscript is now with our production department. 

Kind regards, 

on behalf of

Dr Peter James Hills 

Academic Editor

PLOS ONE